# Semi-supervised batch learning from logged data

## Abstract

Offline policy learning methods are intended to learn a policy from logged data, which includes context, action, and reward for each sample point. In this work we build on the counterfactual risk minimization framework, which also assumes access to propensity scores. We propose learning methods for problems where rewards of some samples are missing, so there are samples with rewards and samples missing rewards in the logged data. We refer to this type of learning as semi-supervised batch learning from logged data, which arises in a wide range of application domains. We derive new upper bound for the true risk under inverse propensity score estimation to better address this kind of learning problem. Using this bound, we propose a regularized semi-supervised batch learning method with logged data where the regularization term is reward-independent and, as a result, can be evaluated using the logged missing-reward data. Consequently, even though reward feedback is only present for some samples, a parameterized policy can be learned by leveraging the missing-reward samples. The results of experiments derived from benchmark datasets indicate that these algorithms achieve policies with better performance in comparison with logging policies.

## 1 Introduction

Offline policy learning from logged data is an important problem in reinforcement learning theory and practice. The logged 'known-rewards' dataset represents interaction logs of a system with its environment; recording context, action, propensity score (i.e., probability of the action selection for a given context under the logging policy), and reward feedback. The literature has considered this setting concerning contextual bandits and partially labeled observations. It is used in many real applications, e.g., recommendation systems (Aggarwal, 2016; Li et al., 2011), personalized medical treatments (Kosorok & Laber, 2019; Bertsimas et al., 2017) and personalized advertising campaigns (Tang et al., 2013; Bottou et al., 2013). However, there are two main obstacles to learning from this kind of logged data: first, the observed reward is available for the chosen action only; and second, the logged data is taken under the logging policy so that it could be biased. Batch learning with logged bandit feedback ( a.k.a. Counterfactual Risk Minimization (CRM) ) is a strategy for off-policy learning from logged 'known-rewards' datasets, which has been proposed by Swaminathan & Joachims (2015a) to tackle these challenges.

Batch learning with logged bandit feedback has led to promising results in some settings, including advertising and recommendation systems. However, there are some scenarios where the logged dataset is generated in an uncontrolled manner, posing significant obstacles such as unobserved rewards for some chosen context and action pairs. For example, consider an advertising system server where some ads (actions) are shown to different clients (contexts) according to a conditional probability (propensity score). Now, suppose that the connections between the clients and the server are corrupted momentarily such that the server does not receive any reward feedback, i.e., whether or not the user has clicked on some ads. Under this scenario, we have access to 'missing-rewards' data indicating the chosen clients, the shown ads, the probability of shown ads but missing reward feedback, and some logged data containing reward feedback. Likewise, there are other scenarios where obtaining reward samples for some context and action (and propensity score) samples may be challenging since it might be expensive or unethical, such as in finance (Musto et al., 2015) or healthcare (Chakrabortty & Cai, 2018).

We call Semi-supervised Batch Learning (S2BL) our approach to learning in these scenarios, where we have access to the logged missing-rewards (no recorded rewards) dataset, besides the logged known-rewards dataset, which was the typical data considered in previous approaches.

This paper proposes algorithms that leverage the logged missing-rewards and known-rewards datasets in an off-policy optimization problem. The contributions of our work are as follows:

- We propose a novel upper bound on the true risk under the inverse propensity score (IPS) estimator in terms of different divergences, including KL divergence and reverse KL divergence between the logging policy and a parameterized learned policy.

- Inspired by this upper bound, we propose regularization approaches based on KL divergence or reverse KL divergence between the logging policy and a parameterized learned policy, which are independent of rewards and hence can be optimized using the logged missing-reward dataset. We also propose a consistent and asymptotically unbiased estimator of KL divergence and reverse KL divergence between the logging policy and a parameterized learned policy.

- We introduce a set of experiments conducted on appropriate datasets to assess the effectiveness of our proposed algorithms. The results demonstrate the adaptability of our approaches in leveraging logged missing-reward data across different configurations, encompassing both linear and deep structures. Furthermore, we offer a comparative analysis against established baselines in the literature.

## 2    RELATED WORKS

There are various methods that have been developed to learn from logged known-reward datasets. The main approach is batch learning with a logged known-reward dataset (bandit feedback), discussed next. We also discuss below some works on the importance weighting, the PAC-Bayesian approach. Appendix A discusses other related topics and the corresponding literature.

**Batch Learning with Logged known-reward dataset:** The mainstream approach for off-policy learning from a logged known-reward dataset is CRM  (Swaminathan & Joachims, 2015a).  In particular, Joachims et al. (2018) proposed a new approach to train a neural network, where the output of the softmax layer is considered as the policy, and the network is trained using the available logged known-reward dataset. Our work builds on the former, albeit proposing methods to learn from logged missing-reward data besides the logged known-reward dataset. CRM has also been combined with domain adversarial networks by Atan et al. (2018). Wu & Wang (2018) proposed a new framework for CRM based on regularization by Chi-square divergence between the parameterized learned policy and the logging policy, and a generative-adversarial approach is proposed to minimize the regularized empirical risk using the logged known-reward dataset. Xie et al. (2018) introduced the surrogate policy method in CRM. The combination of causal inference and counterfactual learning was studied by Bottou et al. (2013). Distributional robust optimization is applied in CRM by Faury et al. (2020). A lower bound on the expected reward in CRM under Self-normalized Importance Weighting was derived by Kuzborskij et al. (2021). The sequential CRM where the loggged known-reward dataset is collected at each iteration of training is studied in Zenati et al. (2023). In this work, we introduce a novel algorithm that leverages both the logged missing-reward dataset and the logged known-reward dataset.

**Importance Weighting:** This method has been proposed for off-policy estimation and learning (Thomas et al., 2015; Swaminathan & Joachims, 2015a). Due to its large variance in many cases (Rosenbaum & Rubin, 1983), some truncated importance sampling methods are proposed, including the IPS estimator with a truncated ratio of policy and logging policy (Ionides, 2008), IPS estimator with truncated propensity score (Strehl et al., 2010) or self-normalizing estimator (Swaminathan & Joachims, 2015b). A balance-based weighting approach for policy learning, which outperforms other estimators, was proposed by Kallus (2018). A generalization of importance sampling by considering samples from different policies is studied by Papini et al. (2019). The weights can be estimated directly by sampling from contexts and actions using Direct Importance Estimation (Sugiyama et al., 2007). A convex surrogate for the regularized true risk by the entropy of parameterized policy is proposed in Chen et al. (2019). An exponential smoothed version of the IPS estimator is proposed by Aouali et al. (2023). Other corrections of IPS estimator are also proposed by Metelli et al. (2021); Su et al. (2020). This work considers the IPS estimator based on a truncated propensity score.

## 3 PRELIMINARIES

**Notations:** We adopt the following convention for random variables and their distributions in the sequel. A random variable is denoted by an upper-case letter (e.g., $Z$), an arbitrary value of this variable is denoted with the lower-case letter (e.g., $z$), and its space of all possible values with the corresponding calligraphic letter (e.g., $\mathcal{Z}$). This way, we can describe generic events like $\{Z = z\}$ for any $z \in \mathcal{Z}$, or events like $\{g(Z) \leq 5\}$ for functions $g : \mathcal{Z} \to \mathbb{R}$. $P_Z$ denotes the probability distribution of the random variable $Z$. The joint distribution of a pair of random variables $(Z_1, Z_2)$ is denoted by $P_{Z_1, Z_2}$. We denote the set of integer numbers from 1 to $n$ by $[n] \triangleq \{1, \cdots, n\}$.

**Divergence Measures:** If $P$ and $Q$ are probability measures over $\mathcal{Z}$, the Kullback-Leibler (KL) divergence $\mathrm{KL}(P\|Q)$ is given by $\mathrm{KL}(P\|Q) \triangleq \int_{\mathcal{Z}} \log\left(\frac{dP}{dQ}\right)dP$ when $P$ is absolutely continuous[1] with respect to $Q$, and $\mathrm{KL}(P\|Q) \triangleq \infty$ otherwise.

The so-called 'reverse KL divergence' is $\mathrm{KL}(Q\|P)$, with arguments in the reverse order. The chi-square divergence is $\chi^2(P\|Q) \triangleq \int_{\mathcal{Z}} (\frac{dP}{dQ})^2 dQ - 1$. For a pair of random variables $(T, Z)$, the conditional KL divergence $\mathrm{KL}(P_{T|Z}\|Q_{T|Z})$ is defined as

$$\mathrm{KL}(P_{T|Z}\|Q_{T|Z}) \triangleq \int_{\mathcal{Z}} \mathrm{KL}(P_{T|Z=z}\|Q_{T|Z=z})dP_Z(z).$$

The conditional chi-square divergence $\chi^2(P_{T|Z}\|Q_{T|Z})$ is defined similarly.

**Problem Formulation:** Let $\mathcal{X}$ be the set of contexts and $\mathcal{A}$ the finite set of actions, with $|\mathcal{A}| = k \geq 2$. We consider policies as conditional distributions over actions, given contexts. For each pair of context and action $(x, a) \in \mathcal{X} \times \mathcal{A}$ and policy $\pi \in \Pi$, where $\Pi$ is the set of policies, the value $\pi(a|x)$ is defined as the conditional probability of choosing action $a$ given context $x$ under the policy $\pi$.

A reward function $r : \mathcal{X} \times \mathcal{A} \to [-1, 0]$, which is unknown, defines the reward of each observed pair of context and action. However, in a *logged known-reward* setting, we only observe the reward (feedback) for the chosen action $a$ in a given context $x$, under the logging policy $\pi_0(a|x)$. We have access to the logged known-reward dataset $S = (x_i, a_i, p_i, r_i)_{i=1}^n$ where each 'data point' $(x_i, a_i, p_i, r_i)$ contains the context $x_i$ which is sampled from unknown distribution $P_X$, the action $a_i$ which is sampled from (partially unknown) logging policy $\pi_0(\cdot|x_i)$, the propensity score $p_i \triangleq \pi_0(a_i|x_i)$, and the observed reward $r_i \triangleq r(x_i, a_i)$ under logging policy $\pi_0(a_i|x_i)$.

The *true risk* of a policy $\pi_\theta$ is,

$$R(\pi_\theta) = \mathbb{E}_{P_X}[\mathbb{E}_{\pi_\theta(A|X)}[r(A, X)]]. \tag{1}$$

Our objective is to find an optimal $\pi_\theta^\star$ which minimizes $R(\pi_\theta)$, i.e., $\pi_\theta^\star = \arg\min_{\pi_\theta \in \Pi_\theta} R(\pi_\theta)$, where $\Pi_\theta$ is the set of all policies parameterized by $\theta \in \Theta$. We denote the importance weighted reward function as $w(A, X)r(A, X)$, where

$$w(A, X) = \frac{\pi_\theta(A|X)}{\pi_0(A|X)}.$$

As discussed by Swaminathan & Joachims (2015b), we can apply the IPS estimator over logged known-reward dataset $S$ (Rosenbaum & Rubin, 1983) to get an unbiased estimator of the risk (an *empirical risk*) by considering the importance weighted reward function as,

$$\hat{R}(\pi_\theta, S) = \frac{1}{n}\sum_{i=1}^n r_i w(a_i, x_i), \tag{2}$$

where $w(a_i, x_i) = \frac{\pi_\theta(a_i|x_i)}{\pi_0(a_i|x_i)}$. The IPS estimator as an unbiased estimator has bounded variance if the $\pi_\theta(A|X)$ is absolutely continuous with respect to $\pi_0(A|X)$ (Strehl et al., 2010; Langford et al., 2008). For the issue of the large variance of the IPS estimator, many estimators are proposed (Strehl et al., 2010; Ionides, 2008; Swaminathan & Joachims, 2015b), e.g., truncated IPS estimator. In this work we consider truncated IPS estimator with threshold $\nu \in (0, 1]$ as follows:

$$\hat{R}_\nu(\pi_\theta, S) = \frac{1}{n}\sum_{i=1}^n r_i w_\nu(a_i, x_i), \tag{3}$$

---

[1]$P$ is absolutely continuous with respect to $Q$ if $P(A) = 0$ whenever $Q(A) = 0$, for measurable $A \subset \mathcal{X}$.

where $w_\nu(a_i, x_i) = \frac{\pi_\theta(a_i, x_i)}{\max(\nu, \pi_0(a_i, x_i))}$. Note that the truncation threshold $\nu \in (0, 1]$ implies an upper bound on the importance weights, $\sup_{(x,a) \in \mathcal{X} \times \mathcal{A}} w_\nu(a, x) \leq \nu^{-1}$.

In our S2BL setting, we also have access to the logged missing-reward dataset, which we shall denote as $S_u = (x_j, a_j, p_j)_{j=1}^m$ and assume it is generated under the same logging policy for the logged known-reward dataset, i.e., $p_j = \pi_0(a_j|x_j)$. We will next propose two algorithms to learn a policy that minimizes the true risk using logged missing-reward and known-reward datasets.

## 4 BOUNDS ON TRUE RISK OF IPS ESTIMATOR

In this section, we provide an upper bound on the variance of importance weighted reward, i.e.,

$$\text{Var}(w(A,X)r(A,X)) \triangleq \mathbb{E}_{P_X \otimes \pi_0(A|X)}\left[(w(A,X)r(A,X))^2\right] - R(\pi_\theta)^2, \quad (4)$$

where $R(\pi_\theta) = \mathbb{E}_{P_X \otimes \pi_0(A|X)}[w(A,X)r(A,X)] = \mathbb{E}_{P_X \otimes \pi_\theta(A|X)}[r(A,X)]$.

Throughout this section we use the simplified notations $\text{KL}(\pi_\theta\|\pi_0) = \text{KL}(\pi_\theta(A|X)\|\pi_0(A|X))$ and $\text{KL}(\pi_0\|\pi_\theta) = \text{KL}(\pi_0(A|X)\|\pi_\theta(A|X))$. All the proofs are deferred to the Appendix C.

**Proposition 1.** *Suppose that the importance weighted of squared reward function, i.e., $w(A,X)r^2(A,X)$, is $\sigma$-sub-Gaussian[2] under $P_X \otimes \pi_0(A|X)$ and $P_X \otimes \pi_\theta(A|X)$, and the reward function has bounded range $[c, b]$ with $b \geq 0$. Then, the following upper bound holds on the variance of the importance weighted reward function:*

$$\text{Var}(w(A,X)r(A,X)) \leq \sqrt{2\sigma^2 \min(\text{KL}(\pi_\theta\|\pi_0), \text{KL}(\pi_0\|\pi_\theta))} + b_u^2 - c_l^2, \quad (5)$$

*where $c_l = \max(c, 0)$ and $b_u = \max(|c|, b)$.*

We have the following Corollary for the truncated IPS estimator with threshold $\nu \in (0, 1]$.

**Corollary 1.** *Assume a bounded reward function with range $[c, 0]$ and a truncated IPS estimator with threshold $\nu \in (0, 1]$. Then the following upper bound holds on the variance of the truncated importance weighted reward function,*

$$\text{Var}(w_\nu(A,X)r(A,X)) \leq c^2(\nu^{-1}\sqrt{\min(\text{KL}(\pi_\theta\|\pi_0), \text{KL}(\pi_0\|\pi_\theta))/2} + 1). \quad (6)$$

Using Cortes et al. (2010, Lemma 1), we can provide an upper bound on the variance of importance weights in terms of the chi-square divergence by considering $r(a, x) \in [c, b]$, as follows:

$$\text{Var}(w(A,X)r(A,X)) \leq b_u^2 \chi^2(\pi_\theta(A|X)\|\pi_0(A|X)) + b_u^2 - c_l^2, \quad (7)$$

where $c_l = \max(c, 0)$ and $b_u = \max(|c|, b)$. In Appendix C.1, we discuss that, under some conditions, the upper bound in Proposition 1 is tighter than the upper bound based on chi-square divergence in (7). The upper bound in Proposition 1 shows that we can reduce the variance of importance weighted reward function, i.e., $w(A,X)r(A,X)$, by minimizing the KL divergence or reverse KL divergence, i.e. $\text{KL}(\pi_\theta\|\pi_0)$ or $\text{KL}(\pi_0\|\pi_\theta)$. A lower bound on the variance of the importance weighted reward function in terms of the KL divergence $\text{KL}(\pi_\theta\|\pi_0)$ is provided in Appendix C.

We can derive a high-probability bound on the true risk under the IPS estimator using the upper bound on the variance of importance weighted reward function in Corollary 1.

**Theorem 1.** *Suppose the reward function takes values in $[-1, 0]$. Then, for any $\delta \in (0, 1)$, the following bound on the true risk of policy $\pi_\theta(A|X)$ with the truncated IPS estimator (with parameter $\nu \in (0, 1]$) holds with probability at least $1 - \delta$ under the distribution $P_X \otimes \pi_0(A|X)$:*

$$R(\pi_\theta) \leq \hat{R}_\nu(\pi_\theta, S) + \frac{2\log(\frac{1}{\delta})}{3\nu n} + \sqrt{\frac{(\nu^{-1}\sqrt{2\min(\text{KL}(\pi_\theta\|\pi_0), \text{KL}(\pi_0\|\pi_\theta))} + 2)\log(\frac{1}{\delta})}{n}}. \quad (8)$$

The proof of Theorem 1 leverages the Bernstein inequality together with an upper bound on the variance of importance weighted reward function using Proposition 1. Theorem 1 shows

---

[2]A random variable $X$ is $\sigma$-subgaussian if $E[e^{\gamma(X-E[X])}] \leq e^{\frac{\gamma^2\sigma^2}{2}}$ for all $\gamma \in \mathbb{R}$.

that we can minimize the KL divergence $\mathrm{KL}(\pi_\theta(A|X)\|\pi_0(A|X))$, or reverse KL divergence $\mathrm{KL}(\pi_0(A|X)\|\pi_\theta(A|X))$, instead of the empirical variance minimization in CRM framework (Swaminathan & Joachims, 2015a) which is inspired by the upper bound in Maurer & Pontil (2009). We compared our upper bound with (London & Sandler, 2019, Theorem 1) in Appendix E.1.

The minimization of KL divergence and reverse KL divergence can also be interpreted from another perspective.

**Proposition 2.** *The following upper bound holds on the absolute difference between risks of logging policy $\pi_0(a|x)$ and the policy $\pi_\theta(a|x)$:*

$$|R(\pi_\theta) - R(\pi_0)| \leq \min\left(\sqrt{\frac{\mathrm{KL}(\pi_\theta\|\pi_0)}{2}}, \sqrt{\frac{\mathrm{KL}(\pi_0\|\pi_\theta)}{2}}\right). \tag{9}$$

Based on Proposition 2, minimizing KL divergence and reverse KL divergence would lead to a policy close to the logging policy in KL divergence or reverse KL divergence. This phenomenon, which is also observed in the works by Swaminathan & Joachims (2015a); Wu & Wang (2018); London & Sandler (2019), is aligned with the fact that the parameterized learned policy should not diverge too much from the logging policy (Schulman et al., 2015). As mentioned by Brandfonbrener et al. (2021) and Swaminathan & Joachims (2015b), the propensity overfitting issues are solved by variance reduction. Therefore, with the KL divergence and reverse KL divergence regularization, we can reduce the propensity overfitting.

## 5 SEMI-SUPERVISED BATCH LEARNING VIA REWARD FREE REGULARIZATION

We now propose our approach for S2BL settings: reward-free regularization. It can leverage the availability of the logged known-reward dataset $S$ and the logged missing-reward dataset $S_u$. The reward-free regularized semi-supervised batch learning is based on optimizing a regularized batch learning objective via logged data, where the regularization function is independent of the rewards. It is inspired by an entropy minimization approach in semi-supervised learning, where one optimizes a label-free entropy function.

Note that the KL divergence $\mathrm{KL}(\pi_\theta\|\pi_0)$ and reverse KL divergence $\mathrm{KL}(\pi_0\|\pi_\theta)$ appearing in Theorem 1 are independent of the reward function values (feedback). This motivates us to consider them as functions that can be optimized using both the logged known-reward and missing-reward datasets. It is worth mentioning that the regularization based on empirical variance proposed by Swaminathan & Joachims (2015a) depends on rewards.

We propose the following truncated IPS estimator regularized by KL divergence $\mathrm{KL}(\pi_\theta\|\pi_0)$ or reverse KL divergence $\mathrm{KL}(\pi_0\|\pi_\theta)$, thus casting S2BL into a semi-supervised CRM problem:

$$\hat{R}_{\mathrm{KL}}(\pi_\theta, S, S_u) \triangleq \hat{R}_\nu(\pi_\theta, S) + \lambda\mathrm{KL}(\pi_\theta(A|X)\|\pi_0(A|X)), \quad \lambda \geq 0, \tag{10}$$

$$\hat{R}_{\mathrm{RKL}}(\pi_\theta, S, S_u) \triangleq \hat{R}_\nu(\pi_\theta, S) + \lambda\mathrm{KL}(\pi_0(A|X)\|\pi_\theta(A|X)), \quad \lambda \geq 0, \tag{11}$$

where for $\lambda = 0$, our problem reduces to traditional batch learning with the logged known-reward dataset that neglects the logged missing-reward dataset. For a large value of $\lambda$, we optimize the KL divergence or reverse KL divergence using both logged missing-reward and known-reward datasets. More discussion for KL regularization is provided in Appendix G.

For the estimation of $\mathrm{KL}(\pi_\theta(A|X)\|\pi_0(A|X))$ and $\mathrm{KL}(\pi_0(A|X)\|\pi_\theta(A|X))$, we can apply the logged missing-reward dataset as follows:

$$\hat{L}_{\mathrm{KL}}(\pi_\theta) \triangleq \sum_{i=1}^{k} \frac{1}{m_{a_i}} \sum_{(x,a_i,p)\in S_u \cup S} \pi_\theta(a_i|x)\log(\pi_\theta(a_i|x)) - \pi_\theta(a_i|x)\log(p), \tag{12}$$

$$\hat{L}_{\mathrm{RKL}}(\pi_\theta) \triangleq \sum_{i=1}^{k} \frac{1}{m_{a_i}} \sum_{(x,a_i,p)\in S_u \cup S} -p\log(\pi_\theta(a_i|x)) + p\log(p), \tag{13}$$

where $m_{a_i}$ is the number of context, action, and propensity score tuples, i.e., $(x,a,p) \in S_u \cup S$, with the same action, e.g., $a = a_i$ (note we have $\sum_{i=1}^{k} m_{a_i} = m + n$). It is possible to show that these estimators of KL divergence and reverse KL divergence are unbiased in the asymptotic sense.

**Proposition 3.** *(proved in Appendix D) Suppose that* $\mathrm{KL}(\pi_\theta(A|X)\|\pi_0(A|X))$ *and the reverse* $\mathrm{KL}(\pi_0(A|X)\|\pi_\theta(A|X))$ *are bounded. Assuming* $m_{a_i} \to \infty$ *(* $\forall a_i \in \mathcal{A}$ *), then* $\hat{L}_{\mathrm{KL}}(\pi_\theta)$ *and* $\hat{L}_{\mathrm{RKL}}(\pi_\theta)$ *are unbiased estimations of* $\mathrm{KL}(\pi_\theta(A|X)\|\pi_0(A|X))$ *and* $\mathrm{KL}(\pi_0(A|X)\|\pi_\theta(A|X))$ *, respectively.*

An estimation error analysis for the proposed estimators in Proposition 3 is conducted in Appendix D. Note that another approach to minimize the KL divergence or reverse KL divergence is $f$-GAN (Wu & Wang, 2018; Nowozin et al., 2016), which is based on using a logged known-reward dataset without considering rewards and propensity scores. It is worthwhile to mention that the generative-adversarial approach will not consider propensity scores in the logged known-reward dataset and also incur more complexity, including Gumbel softmax sampling (Jang et al., 2016) and discriminator network optimization. We proposed a new estimator of these information measures considering our access to propensity scores in the logged missing-reward dataset. Since the term $p\log(p)$ in (13) is independent of policy $\pi_\theta$, we ignore it and optimize the following quantity instead of $\hat{L}_{\mathrm{RKL}}(\pi_\theta, S_u)$ which is similar to cross-entropy by considering propensity scores as weights of cross-entropy:

$$\hat{L}_{\mathrm{WCE}}(\pi_\theta) \triangleq \sum_{i=1}^{k} \frac{1}{m_{a_i}} \sum_{(x,a_i,p) \in S_u \cup S} -p\log(\pi_\theta(a_i|x)). \tag{14}$$

For improvement in regularization with KL divergence in the scenarios where the propensity scores in the logged missing-reward dataset are zero, we use the propensity score truncation in (12) as follows:

$$\hat{L}_{\mathrm{KL}}^{\nu}(\pi_\theta) \triangleq \sum_{i=1}^{k} \frac{1}{m_{a_i}} \sum_{(x,a_i,p) \in S_u \cup S} \pi_\theta(a_i|x)\log\left(\pi_\theta(a_i|x)\right) - \pi_\theta(a_i|x)\log(\max(\nu, p)), \tag{15}$$

where $\nu \in [0,1]$ is the same truncation parameter for truncated IPS estimator in (3). Note that in a case of $p_i = 0$ for some sample $(x_i, a_i, p_i) \in S_u$ then we have $\hat{L}_{\mathrm{KL}} = -\infty$; hence considering $\nu$ in $\hat{L}_{\mathrm{KL}}^{\nu}$ will help to solve these cases.

## 6 ALGORITHMS AND EXPERIMENTS

We briefly present our experiments. More details and discussions can be found in Appendix F. We consider two approaches, softmax policy with linear model inspired by Swaminathan & Joachims (2015a); London & Sandler (2019) and the softmax policy via deep model inspired by Joachims et al. (2018).

**Softmax policy with linear model:** Following the prior works Swaminathan & Joachims (2015a); London & Sandler (2019), we consider the stochastic softmax policy as,

$$\pi_{\tilde{\theta}}(a_i|x) = \frac{\exp(\tilde{\theta}.\phi(a_i, x))}{\sum_{j=1}^{k} \exp(\tilde{\theta}.\phi(a_j, x))}, \tag{16}$$

where $\phi(a_i, x)$ is a feature map for $(a_i, x)$ and $\tilde{\theta}$ is the vector of parameters. Therefore, our parameterized learned policy is based on a linear model.

**Softmax policy with deep model:** Following Joachims et al. (2018), we consider the output of a softmax layer in a neural network as a stochastic parameterized learned policy,

$$\pi_\theta(a_i|x) = \frac{\exp(h_\theta(x, a_i))}{\sum_{j=1}^{k} \exp(h_\theta(x, a_j))}, \tag{17}$$

where $h_\theta(x, a_i)$ is the $i$-th input to softmax layer for context $x \in \mathcal{X}$ and action $a_i \in \mathcal{A}$.

**Baselines:** For linear model, we consider the Bayesian CRM, London & Sandler (2019), as a baseline to compare with our algorithms. More details for comparison of our algorithm with Bayesian CRM in provided in Appendix E.2. For deep model, we consider the BanditNet as a baseline in our experiment. More details regarding the BanditNet is provided in Appendix F.4.

**Algorithms:** The WCE-S2BL algorithm, proposed in Algorithm 1, is based on reward-free regularized truncated IPS estimator in linear model via truncated weighted cross-entropy. The KL-S2BL

algorithm is similar to Algorithm 1 by replacing $\hat{L}_{\text{WCE}}(\theta^{t_g})$ with $\hat{L}_{\text{KL}}^{\nu}(\theta^{t_g})$ defined as,

$$\hat{L}_{\text{KL}}^{\nu}(\theta^{t_g}) = \sum_{i=1}^{k} \frac{1}{m_{a_i}} \sum_{(x,a_i,p) \in S_u \cup S} \pi_{\theta^{t_g}}(a_i|x) \log\left(\frac{\pi_{\theta^{t_g}}(a_i|x)}{\max(\nu,p)}\right). \tag{18}$$

We examine the performance of the algorithms WCE-S2BL and KL-S2BL in both linear and deep models. For a fair comparison, we run experiments for WCE-S2BL and KL-S2BL using the logged known-reward dataset for regularization. These algorithms are referred to as WCE-S2BLK and KL-S2BLK, respectively. Note that in linear model, we have truncated IPS estimator. However, in the deep model, we consider BanditNet which is based on self-normalized IPS estimator. Therefore, in the WCE-S2BL algorithm for deep model we replace the truncated IPS estimator via BanditNet approach Joachims et al. (2018) in Algorithm 1.

---

**Algorithm 1:** WCE-S2BL Algorithm for Linear Model

---

**Data:** $S = (x_i, a_i, p_i, r_i)_{i=1}^{n}$ sampled from $\pi_0$, $S_u = (x_j, a_j, p_j)_{j=1}^{m}$ sampled from $\pi_0$, hyper-parameters $\lambda$ and $\nu$, initial policy $\pi_{\theta^0}(a|x)$, epoch index $t_g$ and max epochs for the whole algorithm $M$

**Result:** An optimized policy $\pi_{\theta}^{\star}(a|x)$ which minimize the regularized risk by weighted cross-entropy

**while** $t_g \leq M$ **do**

     Sample $n$ samples $(x_i, a_i, p_i, r_i)$ from $S$ and estimate the re-weighted loss as

     $\hat{R}_{\nu}(\theta^{t_g}) = \frac{1}{n} \sum_{i=1}^{n} r_i \frac{\pi_{\theta^{t_g}}(a_i|x_i)}{\max(\nu,p_i)}$.

     Get the gradient with respect to $\theta^{t_g}$ as $g_1 \leftarrow \nabla_{\theta^{t_g}} \hat{R}_{\nu}(\theta^{t_g})$.

     Sample $m$ samples from $S_u$ and estimate the weighted cross-entropy loss ($\sum_{i=1}^{k} m_{a_i} = m$).

     $\hat{L}_{\text{WCE}}(\theta^{t_g}) = \sum_{i=1}^{k} \frac{1}{m_{a_i}} \sum_{(x,a_i,p) \in S_u \cup S} -p \log(\pi_{\theta^{t_g}}(a_i|x))$.

     Get the gradient with respect to $\theta^{t_g}$ as $g_2 \leftarrow \nabla_{\theta^{t_g}} \hat{L}_{\text{WCE}}(\theta^{t_g})$.

     Update $\theta^{t_g+1} = \theta^{t_g} - (g_1 + \lambda g_2)$.

     $t_g = t_g + 1$.

**end**

---

**Datasets:** We apply the standard supervised to bandit transformation (Beygelzimer & Langford, 2009) on two image classification datasets: Fashion-MNIST (FMNIST) (Xiao et al., 2017) and CIFAR-10 (Krizhevsky, 2009). In Appendix F, we also consider other datasets, including CIFAR-100 and EMNIST. This transformation assumes that each of the ten classes in the datasets corresponds to an action. Then, a logging policy stochastically selects an action for every sample in the dataset. For each data sample $x$, action $a$ is sampled by logging policy. For the selected action, propensity score $p$ is determined by the softmax value of that action. If the selected action matches the actual label assigned to the sample, then we have $r = -1$, and $r = 0$ otherwise. So, the 4-tuple $(x, a, p, r)$ makes up the dataset.

**Logging policy:** To learn logging policies with different performances, given inverse temperature[3] $\tau \in \{1, 5, 10, 20\}$ we train a simplified ResNet architecture having a single residual layer in each block with inverse temperature $\tau$ in the softmax layer on the fully-labeled dataset, FMNIST and CIFAR-10. Then, we augment the dataset with the outputs and rewards of the trained policy, this time with inverse temperature equal to $1$ in the softmax layer. Hence, the learned policy is logged with inverse temperature $\tau$. Increasing $\tau$ leads to more uniform and less accurate logging policies.

We evaluate the performance of the different algorithms based on the accuracy of the trained model. Inspired by London & Sandler (2019), we calculate the accuracy for deterministic policy where the accuracy of model based on the argmax of the softmax layer output for a given context is computed.

To simulate the absence of rewards for logged missing-reward datasets, we pretended that the reward was only available in $\rho \in \{0.02, 0.2\}$ of the samples in each dataset, while the reward of the remaining

---

[3] The inverse temperature $\tau$ is defined as $\pi_0(a_i|x) = \frac{\exp(h(x,a_i)/\tau)}{\sum_{j=1}^{k} \exp(h(x,a_j)/\tau)}$ where $h(x, a_i)$ is the $i$-th input to the softmax layer for context $x \in \mathcal{X}$ and action $a_i \in \mathcal{A}$.

samples is missed. Recall that the regularization term is minimized via both logged known-reward and logged missing-reward datasets.

For each value of $\tau$ and $\rho$ and for both types of deep and linear models, we apply WCE-S2BL, KL-S2BL, WCE-S2BLK and KL-S2BLK, and observe the accuracy over three runs. Figure 2 shows the accuracy of WCE-S2BL, WCE-S2BLK, KL-S2BL and KL-S2BLK methods compared to BanditNet Joachims et al. (2018) for the deep model approach, for $\tau = 10$ and different number of known-reward samples, in the FMNIST and CIFAR-10 datasets. The error bars represent the standard deviation over the three runs. Figure 1 shows similar results for the linear model. Table 2 shows the deterministic accuracy of WCE-S2BL, KL-S2BL, WCE-S2BLK, KL-S2BLK and BanditNet methods for $\tau \in \{1, 10\}$, and $\rho \in \{0.02, 0.2\}$. More results for other values of $\tau$ and $\rho$ are available in Appendix F.5. More experiments about the effect of logged missing-reward dataset and the minimization of regularization terms are available at Appendix F.9.

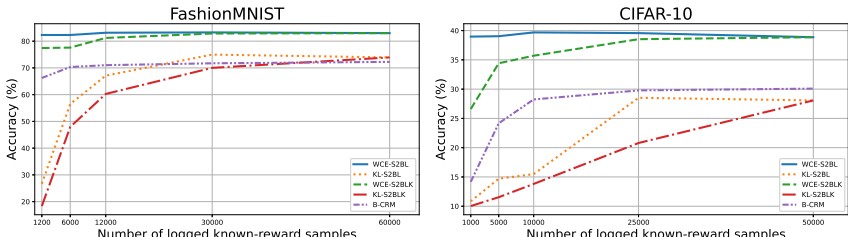

Figure 1: Accuracy of WCE-S2BL, KL-S2BL,WCE-S2BLK, KL-S2BLK, and B-CRM for $\tau = 10$.

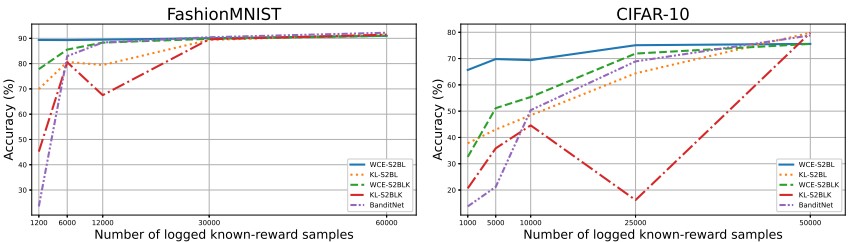

Figure 2: Accuracy of WCE-S2BL, KL-S2BL,WCE-S2BLK, KL-S2BLK, and BanditNet for $\tau = 10$.

Table 1: Comparison of different algorithms WCE-S2BL, KL-S2BL, WCE-S2BLK, KL-S2BLK and Bayesian-CRM (B-CRM) deterministic accuracy for FMNIST and CIFAR-10 with linear layer setup and for different qualities of logging policy ($\tau \in \{1, 10\}$) and proportions of labeled data ($\rho \in \{0.02, 0.2\}$).

| Dataset | $\tau$ | $\rho$ | **WCE-S2BL** | **KL-S2BL** | **WCE-S2BLK** | **KL-S2BLK** | **B-CRM** | **Logging Policy** |
|---|---|---|---|---|---|---|---|---|
| FMNIST | 1 | 0.02 | $84.37 \pm 0.14$ | $71.67 \pm 0.26$ | $78.84 \pm 0.05$ | $74.71 \pm 0.06$ | $64.67 \pm 1.44$ | **91.73** |
| | | 0.2 | $83.59 \pm 0.18$ | $71.88 \pm 0.31$ | $83.05 \pm 0.06$ | $74.06 \pm 0.00$ | $70.99 \pm 0.32$ | |
| | 10 | 0.02 | $\mathbf{82.31 \pm 0.07}$ | $26.71 \pm 2.18$ | $77.43 \pm 0.13$ | $18.35 \pm 7.06$ | $66.24 \pm 00.03$ | 20.72 |
| | | 0.2 | $\mathbf{83.15 \pm 0.09}$ | $67.10 \pm 5.17$ | $81.20 \pm 0.12$ | $60.26 \pm 0.88$ | $71.02 \pm 0.30$ | |
| CIFAR-10 | 1 | 0.02 | $39.39 \pm 0.15$ | $37.21 \pm 0.15$ | $30.56 \pm 0.61$ | $30.08 \pm 0.27$ | $19.00 \pm 1.77$ | **79.77** |
| | | 0.2 | $40.66 \pm 0.29$ | $37.88 \pm 0.58$ | $38.22 \pm 0.01$ | $35.70 \pm 0.25$ | $29.32 \pm 0.35$ | |
| | 10 | 0.02 | $38.97 \pm 0.03$ | $10.84 \pm 1.18$ | $26.60 \pm 0.89$ | $10.03 \pm 0.05$ | $14.17 \pm 2.87$ | **43.45** |
| | | 0.2 | $39.69 \pm 0.05$ | $15.49 \pm 2.23$ | $35.71 \pm 0.73$ | $13.81 \pm 2.74$ | $28.24 \pm 0.20$ | |

Our methods achieves maximum accuracy even when the logging policy's accuracy is not well. For example, in Tables 2 for the CIFAR-10 in deep model setup with $\tau = 10$ and $\rho = 0.2$, we observe $\mathbf{69.40 \pm 0.47}$ accuracy for WCE-S2BL in comparison with $\mathbf{50.38 \pm 0.55}$ and $\mathbf{43.45}$ for BanditNet and logging policy, respectively. We also run more experiments in Appendix F.9 to investigate the effect of logged missing-reward dataset when size of logged known-reward dataset is fixed.

**Discussion:** In most cases, as shown in Tables 2 and 1 (also the extra experiments in Appendix F), WCE-S2BL can achieve a better policy and preserve a more stable behavior compared to baselines and the logging policy in both scenarios, linear and deep learning if we have access to both logged missing-reward dataset and logged known-reward dataset. In KL-S2BL, which employs the KL regularization,

Table 2: Comparison of different algorithms WCE-S2BL, KL-S2BL, WCE-S2BLK, KL-S2BLK and BanditNet deterministic accuracy for FMNIST and CIFAR-10 with deep model setup and different qualities of logging policy ($\tau \in \{1, 10\}$) for different proportions of labeled data ($\rho \in \{0.02, 0.2\}$).

| Dataset | $\tau$ | $\rho$ | WCE-S2BL | KL-S2BL | WCE-S2BLK | KL-S2BLK | BanditNet | Logging Policy |
|---------|--------|--------|----------|---------|-----------|----------|-----------|----------------|
| FMNIST | 1 | 0.2 | **93.16 ± 0.18** | 92.04 ± 0.13 | 82.76 ± 4.45 | 87.72 ± 0.53 | 89.60 ± 0.49 | 91.73 |
| | | 0.02 | **93.12 ± 0.16** | 91.79 ± 0.16 | 78.66 ± 0.90 | 61.46 ± 9.97 | 78.64 ± 1.97 | 91.73 |
| | 10 | 0.2 | **89.47 ± 0.3** | 79.45 ± 0.75 | 88.31 ± 0.14 | 67.53 ± 2.06 | 88.35 ± 0.45 | 20.72 |
| | | 0.02 | **89.35 ± 0.15** | 69.94 ± 0.60 | 77.82 ± 0.73 | 45.18 ± 19.82 | 23.52 ± 3.15 | 20.72 |
| CIFAR-10 | 1 | 0.2 | 85.06 ± 0.32 | **85.53 ± 0.56** | 58.04 ± 5.47 | 54.12 ± 0.51 | 67.96 ± 0.62 | 79.77 |
| | | 0.02 | **85.01 ± 0.37** | 84.60 ± 0.65 | 17.12 ± 0.97 | 21.63 ± 1.44 | 27.39 ± 3.47 | 79.77 |
| | 10 | 0.2 | **69.40 ± 0.47** | 48.44 ± 0.26 | 55.38 ± 3.63 | 44.60 ± 0.19 | 50.38 ± 0.55 | 43.45 |
| | | 0.02 | **65.67 ± 1.06** | 37.80 ± 0.85 | 32.61 ± 1.14 | 20.66 ± 5.74 | 13.78 ± 1.99 | 43.45 |

$KL(\pi_\theta \| \pi_0)$, the policy, $\pi_\theta$, must be absolutely continuous with respect to the logging policy, $\pi_0$. Thus, if the logging policy is zero at an optimal action for a given context, the parameterized learned policy cannot explore this action for the given context. Conversely, in WCE-S2BL, which uses the reverse KL regularization, $KL(\pi_0 \| \pi_\theta)$, the logging policy has to be absolutely continuous with respect to the parameterized learned policy. Hence, when the logging policy is zero at an optimal action for a given context, the reverse KL regularization minimization framework provides an opportunity to explore this action for the given context and have more robust behaviour. It's notable that by minimizing WCE-S2BL and KL-S2BL using only the logged known-reward dataset (introduced as WCE-S2BLK and KL-S2BLK, respectively), we can observe improved performance with respect to the baselines in the most of experiments. This indicates that our regularization is also applicable even when exclusively using a logged known-reward dataset. More discussions are provided in Appendix F.8.

## 7 CONCLUSION AND FUTURE WORKS

We proposed an algorithm, namely, reward-free regularized truncated IPS estimator, for Semi-supervised Batch Learning (S2BL) with logged data settings, effectively casting these kinds of problems into semi-supervised batch learning problems with logged known-reward and missing-reward dataset. The main take-away in reward-free regularized batch learning is proposing regularization terms, i.e., KL divergence and reverse KL divergence between logging policy and parameterized learned policy, independent of reward values, and also the minimization of these terms results in a tighter upper bound on true risk. Experiments revealed that in most cases, these algorithms can reach a parameterized learned policy performance superior to the partially unknown logging policy by exploiting the logged missing-reward dataset and logged known-reward dataset.

The main limitation of this work is the assumption of access to a clean propensity score relating to the probability of an action given a context under the logging policy. We also use propensity scores in both the main objective function and the regularization term. However, we can estimate the propensity score using different methods, e.g., logistic regression (D'Agostino Jr, 1998; Weitzen et al., 2004), generalized boosted models (McCaffrey et al., 2004), neural networks (Setoguchi et al., 2008), parametric modeling (Xie et al., 2019) or classification and regression trees (Lee et al., 2010; 2011). Note that, as discussed in Tsiatis (2006); Shi et al. (2016), under the estimated propensity scores, the variance of IPS estimator reduces. Therefore, a future line of research is to investigate how different methods of propensity score estimation can be combined with our algorithm to optimize the expected risk using logged known-reward and missing-reward datasets. Likewise, we believe that the idea of KL-S2BL and WCE-S2BL can be extended to semi-supervised reward learning and using unlabeled data scenarios in reinforcement learning (Konyushkova et al., 2020; Yu et al., 2022). We can also apply KL-S2BL and WCE-S2BL to other corrections of IPS estimators, (Dudík et al., 2011; Su et al., 2020; Metelli et al., 2021; Aouali et al., 2023) in order to utilize the logged missing-reward dataset. As our current theoretical results hold for truncated IPS estimator, it would be interesting to investigate the effect of our proposed regularization methods on the variance of self-normalized IPS which is used in BanditNet approach.

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

# A OTHER RELATED WORKS

In this section, we discuss more related works about direct methods, inverse reinforcement learning, individualized treatment effects, regularized reinforcement learning with KL divergence, semi-supervised learning, semi-supervised reinforcement learning, causal inference with missing outcomes and PAC-Bayesian approach.

**Direct Method:** The direct method for off-policy learning from logged known-reward datasets is based on the estimation of the reward function, followed by the application of a supervised learning algorithm to the problem (Dudík et al., 2014). However, this approach fails to generalize well, as shown by Beygelzimer & Langford (2009). Another direct-oriented method for off-line policy learning, using the self-training approaches in semi-supervised learning, was proposed by Gao et al. (2022). A different approach based on policy optimization and boosted base learner is proposed to improve the performance in direct methods London et al. (2023). Our approach differs from this area, as the reward function is not estimated and is based on semi-supervised batch learning with logged known-reward and missing-reward datasets.

**Inverse Reinforcement Learning:** Inverse RL, which aims to learn reward functions in a data-driven manner, has also been proposed for the setting of missing-reward datasets in RL (Finn et al., 2016; Konyushkova et al., 2020; Abbeel & Ng, 2004). The identifiability of reward function learning under entropy regularization is studied by Cao et al. (2021). Our work differs from this line of research, since we assume access to propensity score parameters, besides the context and action. Our logged known-reward and missing-reward datasets are under a fixed logging policy for all samples.

**Individualized Treatment Effects:** The individual treatment effect aims to estimate the expected values of the squared difference between outcomes (rewards) for control and treated contexts (Shalit et al., 2017). In the individual treatment effect scenario, the actions are limited to two actions (treated/not treated) and the propensity scores are unknown (Shalit et al., 2017; Johansson et al., 2016; Alaa & van der Schaar, 2017; Athey et al., 2019; Shi et al., 2019; Kennedy, 2020; Nie & Wager, 2021).Recently, the average treatment effects in semi-supervised settings (a.k.a. limited outcome data) from causal (or non-causal) inference perspective is studied by (Zhang et al., 2023b; Chakrabortty et al., 2022; Kallus & Mao, 2020). Our work differs from this line of works by considering larger action spaces and assuming the access to propensity scores for logged datasets.

**Regularized Reinforcement Learning with KL Divergence:** The KL divergence regularization with a logging policy and another parameterized learned policy is studied in off-policy reinforcement learning Wu et al. (2019); Levine et al. (2020); Rudner et al. (2021); Jaques et al. (2019). Our work differs from this line of work by considering a counterfactual risk minimization framework. Our datasets also contain propensity scores, which are unavailable in off-policy reinforcement learning.

**Semi-Supervised Learning:** There are some connections between our scenario, and semi-supervised learning (Yang et al., 2021) approaches, including entropy minimization and pseudo-labeling. In entropy minimization, an entropy function of predicted conditional distribution is added to the main empirical risk function, which depends on unlabeled data (Grandvalet & Bengio, 2005). The entropy function can be viewed as an entropy regularization and can lower the entropy of prediction on unlabeled data. In Pseudo-labeling, the model is trained using labeled data in a supervised manner and is also applied to unlabeled data in order to provide a pseudo label with high confidence (Lee et al., 2013). These pseudo-labels would be applied as inputs for another model, trained based on labeled and pseudo-label data in a supervised manner. Similar methods have been employed in the statistics literature (see e.g., Chakrabortty & Cai, 2018; Gronsbell & Cai, 2018). Our work differs from the aforementioned semi-supervised learning as the logging policy biases our logged data, and the rewards for actions other than the chosen action are unavailable. In semi-supervised learning, the label is missing for some of the data. In comparison, in our setup, the reward is missing. Note that, inspired by the Pseudo-labeling algorithm in semi-supervised learning and also the work by Konyushkova et al. (2020), we can use a model based on the logged known-reward dataset to assign pseudo-rewards to the logged missing-reward dataset and then the final model is trained using the logged known-reward dataset and logged missing-reward dataset augmented by pseudo-rewards. Note that a regularization to reduce the variance of the IPS estimator can also be added. However, as discussed by Beygelzimer & Langford (2009), the model fails to generalize well in the direct method where we estimate the reward function. Therefore, we do not study this method.

**Semi-Supervised Reinforcement Learning:** There are a few proposals that considered off-policy evaluation from logged data in semi-supervised learning settings from individual treatment effect (Sonabend-W et al., 2020b; Cheng et al., 2021). We target a different problem on offline policy learning. Recently, Sonabend-W et al. (2020a) and Gunn et al. (2022) studied semi-supervised offline policy learning. However, an important aspect overlooked in their proposals is the regularization of the uncertainty associated with the value of the parameterized policy. This omission could potentially lead to sub-optimal policies in settings where specific actions have received limited exploration, a common occurrence in observational datasets (Levine et al., 2020).

**PAC-Bayesian Approach:** Some theoretical works for error analysis in this field are focused on the PAC-Bayesian approach (see Hellström et al. (2023) for a recent review), e.g., London & Sandler (2019); Sakhi et al. (2023); Aouali et al. (2023). London & Sandler (2019) leveraged PAC-Bayesian theory inspired by McAllester (2003) to derive an upper bound on the population risk of the parameterized policy for truncated IPS in terms of the KL divergence, with prior and posterior distributions over the hypothesis space. Tighter generalization upper bounds via PAC-Bayesian approach is proposed by Sakhi et al. (2023). Aouali et al. (2023) also applied the PAC-Bayesian approach to analyze the error of the proposed estimator. In this work, our approach is different from the PAC-Bayesian approach, and we provide an upper bound on the variance of the IPS estimator based on the KL divergence between the parameterized and logging policies.

**Pessimism Method and Off-policy Reinforcement Learning:** The pessimism concept is originally introduced in offline reinforcement learning (RL) (Buckman et al., 2020; Jin et al., 2021), aims to derive an optimal policy within Markov decision processes (MDPs) by utilizing pre-existing datasets (Rashidinejad et al., 2022; 2021; Yin & Wang, 2021; Yan et al., 2023). This concept has also been adapted to contextual bandits, viewed as a specific MDP instance. Recently, a 'design-based' version of the pessimism principle is proposed by (Jin et al., 2022) where the author propose a data-dependent and policy-dependent regularization inspired by a lower confidence bound (LCB) on the estimation uncertainty of the augmented-inverse-propensity-weighted (AIPW)-type estimators which also includes IPS estimators. Our work differs from (Jin et al., 2022), as our regularization is inspired by variance reduction of truncated IPS estimator. However, the regularization in (Jin et al., 2022) is motivated by a LCB. In addition, our regularization, can be implemented by deep neural networks. It is also interesting to apply our method in Proposition 2 to provide a lower confidence bound in pessimistic framework in terms of KL-divergence or reverse-KL divergence.

## B  PRELIMINARIES

**Lemma 1.** *Suppose that $f(X)$ is $\sigma$-sub-Gaussian under distribution $Q_X$. Then, considering the difference of expectations of $f(X)$ with respect to a distribution $P_X$ and the distribution $Q_X$, the following upper bound holds:*

$$|\mathbb{E}_{P_X}[f(X)] - \mathbb{E}_{Q_X}[f(X)]| \leq \sqrt{2\sigma^2 \mathrm{KL}(P_X \| Q_X)} \tag{19}$$

*Proof.* From the Donsker-Varadhan representation of KL divergence (Polyanskiy & Wu, 2014), for $\gamma \in \mathbb{R}$ we have:

$$\mathrm{KL}(P_X \| Q_X) \geq \mathbb{E}_{P_X}[\gamma f(X)] - \log(\mathbb{E}_{Q_X}[e^{\gamma f(X)}]) \tag{20}$$

$$\geq \gamma(\mathbb{E}_{P_X}[f(X)] - \mathbb{E}_{Q_X}[f(X)]) - \frac{\gamma^2 \sigma^2}{2} \tag{21}$$

where (21) is the result of sub-Gaussian assumption. We have:

$$\frac{\gamma^2 \sigma^2}{2} - \gamma(\mathbb{E}_{P_X}[f(X)] - \mathbb{E}_{Q_X}[f(X)]) + \mathrm{KL}(P_X \| Q_X) \geq 0. \tag{22}$$

As in (22), we have a quadratic in $\gamma$, which is positive and has a non-positive discriminant, then the final result holds.  □

## C  PROOFS OF SECTION 4

We first prove the following Lemma:

**Proposition 1.** *(restated) Suppose that the importance weighted of squared reward function, i.e.,* $w(A, X)r^2(A, X)$*, is* $\sigma$*-sub-Gaussian under* $P_X \otimes \pi_0(A|X)$ *and* $P_X \otimes \pi_\theta(A|X)$*, and the reward function has bounded range* $[c, b]$ *with* $b \geq 0$*. Then, the following upper bound holds on the variance of the importance weighted reward function:*

$$\text{Var}\left(w(A, X)r(A, X)\right) \leq \sqrt{2\sigma^2 \min(\text{KL}(\pi_\theta \| \pi_0), \text{KL}(\pi_0 \| \pi_\theta))} + b_u^2 - c_l^2, \tag{23}$$

*where* $c_l = \max(c, 0)$*,* $b_u = \max(|c|, b)$*,* $\text{KL}(\pi_\theta \| \pi_0) = \text{KL}(\pi_\theta(A|X) \| \pi_0(A|X))$ *and* $\text{KL}(\pi_0 \| \pi_\theta) = \text{KL}(\pi_0(A|X) \| \pi_\theta(A|X))$.

*Proof.* Note that $c_l^2 \leq R^2(\pi_\theta) \leq b_u^2$ where $c_l = \max(c, 0)$ and $b_u = \max(|c|, b)$.

$$\text{Var}\left(w(A, X)r(A, X)\right) = \mathbb{E}_{P_X \otimes \pi_0(A|X)}\left[\left(w(A, X)r(A, X)\right)^2\right] - R^2(\pi_\theta) \tag{24}$$

$$\leq \mathbb{E}_{P_X \otimes \pi_0(A|X)}\left[\left(w(A, X)r(A, X)\right)^2\right] - c_l^2 \tag{25}$$

where $c_l = \max(c, 0)$. We need to provide an upper bound on $\mathbb{E}_{P_X \otimes \pi_0(A|X)}\left[\left(w(A, X)r(A, X)\right)^2\right]$. First, we have:

$$\mathbb{E}_{P_X \otimes \pi_0(A|X)}\left[\left(w(A, X)r(A, X)\right)^2\right] = \mathbb{E}_{P_X \otimes \pi_0(A|X)}\left[\left(\frac{\pi_\theta(A|X)}{\pi_0(A|X)}r(A, X)\right)^2\right] \tag{26}$$

$$= \mathbb{E}_{P_X \otimes \pi_\theta(A|X)}\left[\frac{\pi_\theta(A|X)}{\pi_0(A|X)}\left(r(A, X)\right)^2\right] \tag{27}$$

Using Lemma 1 and assuming sub-Gaussianity under $P_X \otimes \pi_0(A|X)$ we have:

$$\left|\mathbb{E}_{P_X \otimes \pi_\theta(A|X)}\left[\frac{\pi_\theta(A|X)}{\pi_0(A|X)}\left(r(A, X)\right)^2\right] - \mathbb{E}_{P_X \otimes \pi_0(A|X)}\left[\frac{\pi_\theta(A|X)}{\pi_0(A|X)}\left(r(A, X)\right)^2\right]\right|$$
$$\leq \sqrt{2\sigma^2 \text{KL}(\pi_\theta(A|X) \| \pi_0(A|X) | P_X)}, \tag{28}$$

and since $r(A, X) \in [c, b]$, we have:

$$\mathbb{E}_{P_X \otimes \pi_0(A|X)}\left[\frac{\pi_\theta(A|X)}{\pi_0(A|X)}\left(r(A, X)\right)^2\right] = \mathbb{E}_{P_X \otimes \pi_\theta(A|X)}\left[\left(r(A, X)\right)^2\right] \leq b_u^2. \tag{29}$$

Considering (29) and (28), the following result holds:

$$\mathbb{E}_{P_X \otimes \pi_\theta(A|X)}\left[\frac{\pi_\theta(A|X)}{\pi_0(A|X)}\left(r(A, X)\right)^2\right] \leq \sqrt{2\sigma^2 \text{KL}(\pi_\theta(A|X) \| \pi_0(A|X))} + b_u^2, \tag{30}$$

By a similar argument and the sub-Gaussianity under $P_X \otimes \pi_\theta(A|X)$, we have:

$$\mathbb{E}_{P_X \otimes \pi_\theta(A|X)}\left[\frac{\pi_\theta(A|X)}{\pi_0(A|X)}\left(r(A, X)\right)^2\right] \leq \sqrt{2\sigma^2 \text{KL}(\pi_0(A|X) \| \pi_\theta(A|X))} + b_u^2, \tag{31}$$

And the final result holds by considering (30), (31), and (26). $\qquad\square$

**Corollary 1.** *(restated) Suppose the reward function has a bounded range* $[c, 0]$ *and a truncated IPS estimator with* $\nu \in (0, 1]$*. Then the following upper bound holds on the variance of the truncated importance weighted reward function:*

$$\text{Var}_{(A,X) \sim \pi_0(A|X) \otimes P_X}\left(w_\nu(A, X)r(A, X)\right) \leq c^2\nu^{-1}\sqrt{\min(\text{KL}(\pi_\theta \| \pi_0), \text{KL}(\pi_0 \| \pi_\theta))/2} + c^2, \tag{32}$$

*where* $w_\nu(A, X) = \frac{\pi_\theta(A, X)}{\max(\nu, \pi_0(A, X))}$*,* $\text{KL}(\pi_\theta \| \pi_0) = \text{KL}(\pi_\theta(A|X) \| \pi_0(A|X))$ *and* $\text{KL}(\pi_0 \| \pi_\theta) = \text{KL}(\pi_0(A|X) \| \pi_\theta(A|X))$.

*Proof.* Define $R_\nu(\pi_\theta) := \mathbb{E}_{(A,X) \sim \pi_0(A|X) \otimes P_X}\left[w_\nu(A, X)r(A, X)\right]$. Note that $0 \leq R_\nu^2(\pi_\theta) \leq c^2$.

$$\text{Var}\left(w(A, X)r(A, X)\right) = \mathbb{E}_{P_X \otimes \pi_0(A|X)}\left[\left(w^\nu(A, X)r(A, X)\right)^2\right] - R_\nu^2(\pi_\theta) \tag{33}$$

$$\leq \mathbb{E}_{P_X \otimes \pi_0(A|X)} \left[ (w^\nu(A,X) r(A,X))^2 \right]. \tag{34}$$

We need to provide an upper bound on $\mathbb{E}_{P_X \otimes \pi_0(A|X)} \left[ (w^\nu(A,X) r(A,X))^2 \right]$. First, we have:

$$\mathbb{E}_{P_X \otimes \pi_0(A|X)} \left[ (w^\nu(A,X) r(A,X))^2 \right] = \mathbb{E}_{P_X \otimes \pi_0(A|X)} \left[ \left( \frac{\pi_\theta(A|X)}{\max(\pi_0(A|X), \nu)} r(A,X) \right)^2 \right] \tag{35}$$

$$\leq \mathbb{E}_{P_X \otimes \pi_\theta(A|X)} \left[ \frac{\pi_\theta(A|X)}{\max(\pi_0(A|X), \nu)} \left( r(A,X) \right)^2 \right]. \tag{36}$$

Using Lemma 1 and the fact that the function

$$0 \leq \frac{\pi_\theta(A|X)}{\max(\pi_0(A|X), \nu)} \left( r(A,X) \right)^2 \leq \frac{c^2}{\nu},$$

is $\frac{c^2}{2\nu}$-sub-Gaussian under any distribution, then we have:

$$\left| \mathbb{E}_{P_X \otimes \pi_\theta(A|X)} \left[ \frac{\pi_\theta(A|X)}{\max(\pi_0(A|X), \nu)} \left( r(A,X) \right)^2 \right] - \mathbb{E}_{P_X \otimes \pi_0(A|X)} \left[ \frac{\pi_\theta(A|X)}{\max(\pi_0(A|X), \nu)} \left( r(A,X) \right)^2 \right] \right|$$

$$\leq \frac{c^2}{\nu\sqrt{2}} \sqrt{\mathrm{KL}(\pi_\theta(A|X) \| \pi_0(A|X))}, \tag{37}$$

and since $r(A,X) \in [c, 0]$, we have:

$$\mathbb{E}_{P_X \otimes \pi_0(A|X)} \left[ \frac{\pi_\theta(A|X)}{\max(\pi_0(A|X), \nu)} \left( r(A,X) \right)^2 \right] = \mathbb{E}_{P_X \otimes \pi_\theta(A|X)} \left[ \left( r(A,X) \right)^2 \right] \leq c^2. \tag{38}$$

Considering (29) and (28), the following result holds:

$$\mathbb{E}_{P_X \otimes \pi_\theta(A|X)} \left[ \frac{\pi_\theta(A|X)}{\pi_0(A|X)} \left( r(A,X) \right)^2 \right] \leq c^2 \nu^{-1} \sqrt{\mathrm{KL}(\pi_\theta(A|X) \| \pi_0(A|X))/2} + c^2. \tag{39}$$

By a similar argument and the sub-Gaussianity under $P_X \otimes \pi_\theta(A|X)$, we have:

$$\mathbb{E}_{P_X \otimes \pi_\theta(A|X)} \left[ \frac{\pi_\theta(A|X)}{\pi_0(A|X)} \left( r(A,X) \right)^2 \right] \leq c^2 \nu^{-1} \sqrt{\mathrm{KL}(\pi_0(A|X) \| \pi_\theta(A|X))/2} + c^2. \tag{40}$$

And the final result holds by considering (30), (31), and (26). $\qquad\square$

We now provide a novel lower bound on the variance of the weighted reward function in the following Proposition.

**Proposition 4.** *(proved in Appendix C) Suppose that $q \leq e^{\mathbb{E}_{P_X \otimes \pi_\theta(A,X)}[\log(|r(A,X)|)]}$, the reward function has bounded range $[c, b]$ with $b \geq 0$, and consider $b_u = \max(|c|, b)$. Then, the following lower bound holds on the variance of importance weighted reward function,*

$$\mathrm{Var}\left( w(A,X) r(A,X) \right) \geq q^2 e^{\mathrm{KL}(\pi_\theta(A|X) \| \pi_0(A|X))} - b_u^2. \tag{41}$$

*Proof.* Note that $c_l^2 \leq R^2(\pi_\theta) \leq b_u^2$ where $c_l = \max(c, 0)$ and $b_u = \max(|c|, b)$.

$$\mathrm{Var}\left( w(A,X) r(A,X) \right) = \mathbb{E}_{P_X \otimes \pi_0(A|X)} \left[ (w(A,X) r(A,X))^2 \right] - R^2(\pi_\theta) \tag{42}$$

$$\geq \mathbb{E}_{P_X \otimes \pi_0(A|X)} \left[ (w(A,X) r(A,X))^2 \right] - b_u^2. \tag{43}$$

First, we have:

$$\mathbb{E}_{P_X \otimes \pi_0(A|X)} \left[ (w(A,X) r(A,X))^2 \right] = \mathbb{E}_{P_X \otimes \pi_0(A|X)} \left[ \left( \frac{\pi_\theta(A|X)}{\pi_0(A|X)} r(A,X) \right)^2 \right] \tag{44}$$

$$= \mathbb{E}_{P_X \otimes \pi_\theta(A|X)} \left[ \frac{\pi_\theta(A|X)}{\pi_0(A|X)} \left( r(A,X) \right)^2 \right]. \tag{45}$$

Considering (45), we provide a lower bound on $\mathbb{E}_{P_X \otimes \pi_\theta(A|X)} \left[ \frac{\pi_\theta(A|X)}{\pi_0(A|X)} \left( r(A, X) \right)^2 \right]$ as follows:

$$\mathbb{E}_{P_X \otimes \pi_\theta(A|X)} \left[ \frac{\pi_\theta(A|X)}{\pi_0(A|X)} \left( r(A, X) \right)^2 \right] = \mathbb{E}_{P_X \otimes \pi_\theta(A|X)} \left[ e^{\log\left( \frac{\pi_\theta(A|X)}{\pi_0(A|X)} \right) + 2\log(|r(A,X)|)} \right] \quad (46)$$

$$\geq e^{\mathbb{E}_{P_X \otimes \pi_\theta(A|X)} \left[ \log\left( \frac{\pi_\theta(A|X)}{\pi_0(A|X)} \right) + 2\log(|r(A,X)|) \right]} \quad (47)$$

$$= e^{\mathrm{KL}(\pi_\theta(A|X)\|\pi_0(A|X))} \left( e^{\mathbb{E}_{P_X \otimes \pi_\theta(A|X)} \left[ \log(|r(A,X)|) \right]} \right)^2$$

$$\geq q^2 e^{\mathrm{KL}(\pi_\theta(A|X)\|\pi_0(A|X))}.$$

Where (47) is based on Jensen-inequality for an exponential function. $\qquad\square$

**Remark 1.** *If we consider $r(a, x) \in [c, b]$ with $b \geq 0$, then we can consider $q = \max(0, c)$.*

The lower bound on the variance of importance weights in Proposition 4 can be minimized by minimizing the KL divergence or reverse KL divergence between $\pi_\theta$ and $\pi_0$.

**Theorem 1.** *(restated) Suppose the reward function takes values in $[-1, 0]$. Then, for any $\delta \in (0, 1)$, the following bound on the true risk of policy $\pi_\theta(A|X)$ with the truncated IPS estimator (with parameter $\nu \in (0, 1]$) holds with probability at least $1 - \delta$ under the distribution $P_X \otimes \pi_0(A|X)$:*

$$R(\pi_\theta) \leq \hat{R}_\nu(\pi_\theta, S) + \frac{2\log(\frac{1}{\delta})}{3\nu n} + \sqrt{\frac{\left( \nu^{-1} \sqrt{2\min(\mathrm{KL}(\pi_\theta\|\pi_0), \mathrm{KL}(\pi_0\|\pi_\theta))} + 2 \right) \log(\frac{1}{\delta})}{n}}, \quad (48)$$

*where $\mathrm{KL}(\pi_\theta\|\pi_0) = \mathrm{KL}(\pi_\theta(A|X)\|\pi_0(A|X))$ and $\mathrm{KL}(\pi_0\|\pi_\theta) = \mathrm{KL}(\pi_0(A|X)\|\pi_\theta(A|X))$.*

*Proof.* Define $R_\nu(\pi_\theta) := \mathbb{E}_{(A,X) \sim \pi_0(A|X) \otimes P_X} \left[ w_\nu(A, X) r(A, X) \right]$. Note that we have $0 \leq R_\nu^2(\pi_\theta) \leq 1$ and

$$R(\pi_\theta) \leq R_\nu(\pi_\theta).$$

Let us consider $Z = \frac{\pi_\theta(A|X)}{\max(\pi_0(A|X), \nu)} r(A, X)$ and $|Z| \leq \nu^{-1}$. Then, we have:

$$\mathrm{Var}(Z) = \mathbb{E}_{P_X \otimes \pi_0(A|X)} \left[ \left( \frac{\pi_\theta(A|X)}{\max(\pi_0(A|X), \nu)} r(A, X) \right)^2 \right] - R_\nu^2(\pi_\theta) \quad (49)$$

$$\leq \nu^{-1} \sqrt{\frac{\min(\mathrm{KL}(\pi_\theta\|\pi_0), \mathrm{KL}(\pi_0\|\pi_\theta))}{2}} + 1,$$

where $\mathrm{KL}(\pi_\theta\|\pi_0) = \mathrm{KL}(\pi_\theta(A|X)\|\pi_0(A|X))$ and $\mathrm{KL}(\pi_0\|\pi_\theta) = \mathrm{KL}(\pi_0(A|X)\|\pi_\theta(A|X))$. Using Bernstein inequality (Boucheron et al., 2013), we also have:

$$Pr\left( R_\nu(\pi_\theta) - \hat{R}_\nu(\pi_\theta, S) > \epsilon \right) \leq \exp\left( \frac{-n\epsilon^2/2}{\mathrm{Var}(Z) + \epsilon\nu^{-1}/3} \right). \quad (50)$$

By setting $\delta = \exp\left( \frac{-n\epsilon^2/2}{\mathrm{Var}(Z) + \epsilon\nu^{-1}/3} \right)$ to match the upper bound in (50) and using the variance upper bound (49), the following upper bound with probability at least $(1 - \delta)$ holds under $P_X \otimes \pi_0(A|X)$:

$$R(\pi_\theta) \leq R_\nu(\pi_\theta) \quad (51)$$

$$\leq \hat{R}_\nu(\pi_\theta, S) + \frac{\nu^{-1} \log(\frac{1}{\delta})}{3n}$$

$$+ \sqrt{\frac{\nu^{-2} \log^2(\frac{1}{\delta})}{9n^2} + \frac{\left( \nu^{-1} \sqrt{2\min(\mathrm{KL}(\pi_\theta\|\pi_0), \mathrm{KL}(\pi_0\|\pi_\theta))} + 2 \right) \log(\frac{1}{\delta})}{n}}, \quad (52)$$

By applying $\sqrt{x + y} \leq \sqrt{x} + \sqrt{y}$ to the last term in (52), the final result holds. $\qquad\square$

**Proposition 2.** *(restated) The following upper bound holds on the absolute difference between risks of logging policy $\pi_0(a|x)$ and the policy $\pi_\theta(a|x)$:*

$$|R(\pi_\theta) - R(\pi_0)| \leq \min\left( \sqrt{\frac{\mathrm{KL}(\pi_\theta\|\pi_0)}{2}}, \sqrt{\frac{\mathrm{KL}(\pi_0\|\pi_\theta)}{2}} \right), \quad (53)$$

*where $\mathrm{KL}(\pi_\theta\|\pi_0) = \mathrm{KL}(\pi_\theta(A|X)\|\pi_0(A|X))$ and $\mathrm{KL}(\pi_0\|\pi_\theta) = \mathrm{KL}(\pi_0(A|X)\|\pi_\theta(A|X))$.*

*Proof.* We have:

$$R(\pi_\theta) = \mathbb{E}_{P_X}[\mathbb{E}_{\pi_\theta(A|X)}[r(A, X)]]. \tag{54}$$

$$R(\pi_0) = \mathbb{E}_{P_X}[\mathbb{E}_{\pi_0(A|X)}[r(A, X)]]. \tag{55}$$

As the reward function is bounded in $[-1, 0]$, then it is $\frac{1}{2}$-sub-Gaussian under all distributions. By considering Lemma 1, the final result holds. □

### C.1 PROPOSITION 1 COMPARISON

Without loss of generality, let us consider $r(a, x) \in [-1, 0]$. For $\sup_{(x,a)\in\mathcal{X}\times\mathcal{A}} \frac{\pi_\theta(a|x)}{\pi_0(a|x)} = \nu^{-1} < \infty$. The upper bound in Corollary 1 by considering the KL divergence $\mathrm{KL}(\pi_\theta\|\pi_0)$ can be written as

$$\mathbb{E}_{P_X\otimes\pi_0(A|X)}\left[\left(\frac{\pi_\theta(A|X)}{\pi_0(A|X)}r(A, X)\right)^2\right] \leq \nu^{-1}\sqrt{\frac{\mathrm{KL}(\pi_\theta(A|X)\|\pi_0(A|X))}{2}} + 1. \tag{56}$$

The upper bound on the second moment of importance weighted reward function in (Cortes et al., 2010, Lemma 1) is as follows:

$$\mathbb{E}_{P_X\otimes\pi_0(A|X)}\left[\left(\frac{\pi_\theta(A|X)}{\pi_0(A|X)}r(A, X)\right)^2\right] \leq \chi^2(\pi_\theta(A|X)\|\pi_0(A|X)) + 1. \tag{57}$$

It is shown by Sason & Verdú (2016) that:

$$D(\pi_\theta(A|X)\|\pi_0(A|X)) \leq \log(\chi^2(\pi_\theta(A|X)\|\pi_0(A|X)) + 1). \tag{58}$$

Using (58) in (56) and comparing to (57), then for $\nu^{-1} < e^2 - 1$, $\exists C \in [0, \nu^{-1}]$, e.g. if $\nu^{-1} = 2$ we have $C \approx 1.28$, where if $\chi^2(\pi_\theta(A|X)\|\pi_0(A|X)) \geq C$, then we have:

$$\log(\chi^2(\pi_\theta(A|X)\|\pi_0(A|X)) + 1) \leq \frac{2(\chi^2(\pi_\theta(A|X)\|\pi_0(A|X)))^2}{\nu^{-2}}. \tag{59}$$

Therefore, the upper bound in Proposition 1 is tighter than (Cortes et al., 2010, Lemma 1) for $\chi^2(\pi_\theta(A|X)\|\pi_0(A|X)) \geq C$ if $\nu^{-1} < e^2 - 1$ and $C$ is the solution of $\log(1 + x) - 2x^2/\nu^{-2} = 0$.

## D PROOFS OF SECTION 5

**Proposition 3.** *(restated) Suppose that $\mathrm{KL}(\pi_\theta(A|X)\|\pi_0(A|X))$ and the reverse $\mathrm{KL}(\pi_0(A|X)\|\pi_\theta(A|X))$ are bounded. Assuming $m_{a_i} \to \infty$ ($\forall a_i \in \mathcal{A}$), then $\hat{L}_{\mathrm{KL}}(\pi_\theta)$ and $\hat{L}_{\mathrm{RKL}}(\pi_\theta)$ are unbiased estimations of $\mathrm{KL}(\pi_\theta(A|X)\|\pi_0(A|X))$ and $\mathrm{KL}(\pi_0(A|X)\|\pi_\theta(A|X))$, respectively.*

*Proof.* First, we have the following decomposition:

$$\mathrm{KL}(\pi_\theta(A|X)\|\pi_0(A|X)) = \sum_{i=1}^{k}\mathbb{E}_{P_X}\left[\left(\pi_\theta(A = a_i|X)\log\left(\frac{\pi_\theta(A = a_i|X)}{\pi_0(A = a_i|X)}\right)\right)\right] \tag{60}$$

$$\mathrm{KL}(\pi_0(A|X)\|\pi_\theta(A|X)) = \sum_{i=1}^{k}\mathbb{E}_{P_X}\left[\pi_0(A = a_i|X)\log\left(\frac{\pi_0(A = a_i|X)}{\pi_\theta(A = a_i|X)}\right)\right]. \tag{61}$$

It suffices to show that:

$$\hat{R}_{\mathrm{KL}}(\pi_\theta) \triangleq \sum_{i=1}^{k}\frac{1}{m_{a_i}}\sum_{(x,a_i,p)\in S_u\cup S}\pi_\theta(a_i|x)\log\left(\frac{\pi_\theta(a_i|x)}{p}\right), \tag{62}$$

$$\hat{R}_{\mathrm{RKL}}(\pi_\theta) \triangleq \sum_{i=1}^{k}\frac{1}{m_{a_i}}\sum_{(x,a_i,p)\in S_u\cup S}-p\log(\pi_\theta(a_i|x)) + p\log(p), \tag{63}$$

As we assume the divergences $\text{KL}(\pi_\theta(A|X)\|\pi_0(A|X))$ and $\text{KL}(\pi_0(A|X)\|\pi_\theta(A|X))$ are bounded, then $\mathbb{E}_{P_X}[\pi_0(a_i|X)\log(\frac{\pi_0(a_i|X)}{\pi_\theta(a_i|X)})]$ and $\mathbb{E}_{P_X}[\pi_\theta(a_i|x)\log(\frac{\pi_\theta(a_i|x)}{\pi_0(a_i|x)})] \; \forall i \in [k]$ exist and they are bounded. Due to the Law of Large Numbers Hsu & Robbins (1947), we have that:

$$\frac{1}{m_{a_i}} \sum_{(x,a_i,p)\in S_u} \pi_0(a_i|x)\log\left(\frac{\pi_0(a_i|x)}{\pi_\theta(a_i|x)}\right) \xrightarrow{m_{a_i}\to\infty} \mathbb{E}_{P_X}\left[\pi_0(a_i|X)\log\left(\frac{\pi_0(a_i|X)}{\pi_\theta(a_i|X)}\right)\right], \quad (64)$$

and

$$\frac{1}{m_{a_i}} \sum_{(x,a_i,p)\in S_u} \pi_\theta(a_i|x)\log\left(\frac{\pi_\theta(a_i|x)}{\pi_0(a_i|x)}\right) \xrightarrow{m_{a_i}\to\infty} \mathbb{E}_{P_X}\left[\pi_\theta(a_i|x)\log\left(\frac{\pi_\theta(a_i|x)}{\pi_0(a_i|x)}\right)\right]. \quad (65)$$

By considering (62), (63) and $m_{a_i} \to \infty$, $\forall i \in [k]$, the final results hold. $\qquad\square$

We also provide an upper bound on the estimation error of the proposed estimator in Proposition 3. Let us define

$$f_{\text{KL}}(x,a) := \pi_\theta(A=a|X=x)\log\left(\frac{\pi_\theta(A=a|X=x)}{\pi_0(A=a|X=x)}\right),$$

and

$$g_{\text{RKL}}(x,a) := \pi_0(A=a|X=x)\log\left(\frac{\pi_0(A=a|X=x)}{\pi_\theta(A=a|X=x)}\right).$$

Note that

$$\mathbb{E}_{P_X}[f_{\text{KL}}(X,a_i)] = \text{KL}(\pi_\theta(A=a_i|X)\|\pi_0(A=a_i|X)),$$

and

$$\mathbb{E}_{P_X}[g_{\text{RKL}}(X,a_i)] = \text{KL}(\pi_0(A=a_i|X)\|\pi_\theta(A=a_i|X)).$$

**Proposition 5.** *Assume that $|f_{\text{KL}}(x,a)| \le B$ and $|g_{\text{RKL}}(x,a)| \le C$ for all $x \in \mathcal{X}$ and $a \in \mathcal{A}$. Then, the following upper bounds hold on estimators of KL divergence and reverse KL divergence in Proposition 3, under distribution $P_X$ with probability at least $1-k\delta$ for $\delta \in (0,1/k]$,*

$$\left|\text{KL}(\pi_\theta(A|X)\|\pi_0(A|X)) - \hat{R}_{\text{KL}}(\pi_\theta)\right| \le B\sqrt{2\log(1/\delta)}\sum_{i=1}^{k}\sqrt{\frac{1}{m_{a_i}}}, \quad (66)$$

*and similarly, we have*

$$\left|\text{KL}(\pi_0(A|X)\|\pi_\theta(A|X)) - \hat{R}_{\text{RKL}}(\pi_\theta)\right| \le C\sqrt{2\log(1/\delta)}\sum_{i=1}^{k}\sqrt{\frac{1}{m_{a_i}}}. \quad (67)$$

*Proof.* From Hoeffding's inequality Boucheron et al. (2013), for each action $a_i \in \mathcal{A}$, the following upper bound holds with probability at least $(1-\delta)$ under distribution $P_X$,

$$\left|\mathbb{E}_{P_X}[f_{\text{KL}}(X,a_i)] - \frac{1}{m_{a_i}}\sum_{j=1}^{m_{a_i}} f_{\text{KL}}(x_j,a_i)\right| \le B\sqrt{\frac{2\log(1/\delta)}{m_{a_i}}}, \quad (68)$$

and similarly

$$\left|\mathbb{E}_{P_X}[g_{\text{RKL}}(X,a_i)] - \frac{1}{m_{a_i}}\sum_{j=1}^{m_{a_i}} g_{\text{RKL}}(x_j,a_i)\right| \le C\sqrt{\frac{2\log(1/\delta)}{m_{a_i}}}. \quad (69)$$

Using the Union bound Vershynin (2018) and considering $|\mathcal{A}| \models k$, the following upper bound holds on the estimation error of the proposed estimator in Proposition 3 under distribution $P_X$ with probability at least $(1-k\delta)$ for $\delta \in (0,1/k]$,

$$\begin{aligned}
&\left|\text{KL}(\pi_\theta(A|X)\|\pi_0(A|X)) - \hat{R}_{\text{KL}}(\pi_\theta)\right| \\
&\le \sum_{i=1}^{k}\left|\text{KL}(\pi_\theta(A=a_i|X)\|\pi_0(A=a_i|X)) - \frac{1}{m_{a_i}}\sum_{j=1}^{m_{a_i}} f_{\text{KL}}(x_j,a_i)\right| \\
&\le B\sqrt{2\log(1/\delta)}\sum_{i=1}^{k}\sqrt{\frac{1}{m_{a_i}}},
\end{aligned} \quad (70)$$

and similarly, we have,

$$
\left| \mathrm{KL}(\pi_0(A|X) \| \pi_\theta(A|X)) - \hat{R}_{\mathrm{RKL}}(\pi_\theta) \right|
$$

$$
\leq \sum_{i=1}^{k} \left| \mathrm{KL}(\pi_0(A = a_i|X) \| \pi_\theta(A = a_i|X)) - \frac{1}{m_{a_i}} \sum_{j=1}^{m_{a_i}} g_{\mathrm{RKL}}(x_j, a_i) \right| \tag{71}
$$

$$
\leq C\sqrt{2\log(1/\delta)} \sum_{i=1}^{k} \sqrt{\frac{1}{m_{a_i}}}.
$$

$\square$

# E  COMPARISON WITH BAYESIAN-CRM

In this section, we compare our work with London & Sandler (2019) from both theoretical and algorithm perspectives.

## E.1  COMPARISON WITH THEOREM 1

We compare our Theorem 1 result with (London & Sandler, 2019, Theorem 1). The upper bound on true risk in (London & Sandler, 2019, Theorem 1) is derived by using the PAC-Bayesian approach, where stochastic policies with action distributions induced by distributions over hypotheses. In particular, the probability of an action $a \in \mathcal{A}$ given a context $x \in \mathcal{X}$, is equal to the probability of a random hypothesis for mapping $h : x \mapsto a$, where the probability of random hypothesis can be induced by prior or posterior distribution, $\mathbb{Q}$ or $\mathbb{P}$.

Suppose that we fix the parameter space for hypotheses set. As discussed, in (London & Sandler, 2019, Section 3.1), if we consider the prior distribution equal to logging policy, then KL divergence $\mathrm{KL}(\mathbb{P}\|\mathbb{Q})$ can be interpreted as $\mathrm{KL}(\pi_\theta\|\pi_0)$. Therefore, we can compare our upper bound in Theorem 1 with (London & Sandler, 2019, Theorem 1) as follows:

- Our upper bound is based on the minimum of KL divergence $D(\pi_\theta(A|X)\|\pi_0(A|X))$ and reverse KL divergence $D(\pi_0(A|X)\|\pi_\theta(A|X))$ and the upper bound in (London & Sandler, 2019, Theorem 1) is based on reverse KL divergence only.

- The upper bound in (London & Sandler, 2019, Theorem 1) has the dominating term with rate $O(\sqrt{\frac{\log(n)}{n}})$ and our upper bound contains a term with rate $O(\frac{1}{\sqrt{n}})$ which dominates the bound.

It is worthwhile to mention that the main advantage of our bound over the PAC-Bayesian is the dependency over the reverse KL divergence, i.e $\mathrm{KL}(\pi_0\|\pi_\theta)$. It helps us to define the WCE-S2BL algorithm based on $\mathrm{KL}(\pi_0\|\pi_\theta)$ as regularization.

## E.2  COMPARISON WITH ALGORITHMS

There are two main methods proposed in London & Sandler (2019).

- **IPS-LPR:** It is inspired by (London & Sandler, 2019, Proposition 1) and the authors propose to minimize the following objective function,

$$
\min_{\theta} \frac{1}{n} \sum_{i=1}^{n} r_i \frac{\pi_\theta(a_i|x_i)}{\max(p_i, \nu)} + \lambda_b \|\theta - \theta_0\|^2, \tag{72}
$$

where $\lambda_b$ is the hyper-parameter and $\theta_0$ is the mean of parameter under prior (logging policy). If we know the logging policy, we can compute the $\theta_0$. Otherwise, we should estimate the mean of logging policy distribution via logged known-reward dataset. The parameterized policy is trained via the logged known-reward dataset. It is worthwhile to mention that in B-CRM, (72), it is assumed that the posterior variance, or variance of parameters $\theta$, is fixed to some small value, e.g., $n^{-1}$. However, in our setup, we directly, estimate the

Table 3: Statistics of the datasets used in our experiments.

| DATA SET | TRAINING SAMPLES | TEST SAMPLES | NUMBER OF ACTIONS | DIMENSION |
|---|---|---|---|---|
| FMNIST | 60000 | 10000 | 10 | $28 \times 28$ |
| EMNIST | 60000 | 10000 | 10 | $28 \times 28$ |
| CIFAR-10 | 50000 | 10000 | 10 | $32 \times 32 \times 3$ |
| CIFAR100 | 50000 | 10000 | 100 | $32 \times 32 \times 3$ |
| KUAIREC | 12,530,806 | 4,676,570 | 10,728 | 1555 |

KL divergence and we have no assumption on variance of parameters. In (72), after the estimation of $\theta_0$, the regularization is similar to $L_2$- regularization of model parameters and it is minimized jointly with the truncated IPS estimator via logged-known reward dataset to derive the parameterized logging policy.

- **WNLL-LPR:** Another algorithm is also proposed in London & Sandler (2019) as WNLL-LPR where the following regularized function would be minimized,

$$\min_{\theta} \frac{1}{n} \sum_{i=1}^{n} r_i \frac{\log(\pi_{\theta}(a_i|x_i))}{\max(p_i, \nu)} + \lambda_b \|\theta - \theta_0\|^2. \tag{73}$$

Note that the main objective function in WNLL-LPR is an upper bound on IPS-LPR as the rewards are non-positive, $r_i \in [-1.0]$. It's also observable that, contrasting with IPS-LPR, which can have negative values, WNLL-LPR remains positive. Therefore, WNLL-LPR is not a tight upper bound. Similarly to IPS-LPR, the regularization is minimized via the logged known-reward dataset after setting $\theta_0$.

## F    EXPERIMENTS

### F.1    SETUP DETAILS

In our experiments, we use the following image classification datasets, Fashion-MNIST (FM-NIST) (Xiao et al., 2017), EMNIST (Cohen et al., 2017), CIFAR-10 and CIFAR-100 (Krizhevsky, 2009). We also use KuaiRec dataset as a real-world example, details explained in section F.7. A summary of the statistics of these datasets is provided in Table 3. We use a combination of manual and automatic hyper-parameter tuning for the learning rate values and regularization coefficient $\lambda$. To be more specific, for the deep model we manually test different hyper-parameters for $\tau = 1, 5$ and use them to set search intervals for other values of $\tau$ and all values of $\tau$ for the deep model. For automatic search we use optuna library. We train each model by 120 and 60 epochs for deep and linear models respectively and use a learning rate multiplier of 0.5 in every 25 epochs. Inspired by BanditNet experiments in Joachims et al. (2018), for the CIFAR-10 dataset, we ignore samples with less than $\nu = 0.001$ propensity score, while for the FMNIST dataset after grid search, we consider $\nu = 0.001$ as the truncation parameter. Table 4 illustrates the experiment settings (Real-world dataset settings are separately in section F.7)

### F.2    DEEP MODEL ARCHITECTURE

We use two simple versions of ResNet architecture. For ResNet-v1 we use a single residual layer in each of the four blocks. For ResNet-v2 we use two residual layers in each of the blocks.

### F.3    BANDIT DATASET GENERATION

We create a bandit dataset consisting of samples $(x, y, a, p, r)$ where $x$ is the context, $y$ is the true label (optimal action), $a$ is the logging policy's action, $p$ is the propensity score, and $r$ is the reward of the action. To do so, starting with a labeled dataset (CIFAR-10, CIFAR-100, EMNIST and FMNIST in our experiments) containing only the pair $(x, y)$ for each sample, we first train a logging policy using the true labels $y$, with fully supervised feedback. For each context $x$ in the labeled dataset, we sample an action $a$ and compute propensity score $p$ from the trained logging policy according

Table 4: Experiment setup details for the softmax policy with deep learning

|  | Deep model | Linear model |
|---|---|---|
| Optimizer | SGD | SGD |
| $\nu$ | 0.001 | 0.001 |
| Network | ResNet-v2 | Linear |
| Learning rate | 0.005 | 0.0005 |
| Max epochs ($M$) | 120 | 60 |
| Batch size | 128 | 128 |

to the softmax output of the model, and compute the reward value $r$. Hence the tuple $(x, y, a, p, r)$ is created. In order to decrease the performance of the logging policy in a controlled manner, for each $\tau \in \{1, 5, 10, 20\}$ we train a logging policy with temperature $\tau$. During dataset generation, we sample from the logging policy with temperature $\tau = 1$. So the trained logging policy's performance decreases as $\tau$ increases. For each $\rho \in \{1, 0.5, 0.2, 0.1, 0.02\}$, we randomly select $\rho$ proportion of the samples and remove the reward from other samples.

Therefore for each labeled dataset, we create $20 = 4 \times 5$ bandit datasets for different values of $\tau$ and $\rho$. For a fair comparison between different methods, we create and store these datasets once, and apply the models on the same dataset for each setting.

For CIFAR-10, FashionMNIST, and EMNIST datasets in linear model, we flatten the image to get a $3072$, $784$, and $784$ dimensional feature vector respectively. For CIFAR-100 we use ResNet-50 pretrained features. We use this vector as the context $x$.

The architecture of the logging policy is ResNet-v1 for CIFAR-10, FMNIST, and EMNIST. For CIFAR-100 we use ResNet-v2.

Note that for linear experiments on CIFAR-10 and FMNIST, we trained a deep logging policy. However for EMNIST and CIFAR-100, we used a linear model for the logging policy. Table 5 shows a summary of features and models in a linear setting. The reason behind the different settings is to observe the difference in performance when the logging policy is of different architectures. We also carried out experiments on CIFAR-10 with linear logging policy, explained in section F.6.

Table 5: Summary of models and features in linear experiments

|  | Logging policy | Trained Policy | Features |
|---|---|---|---|
| FMNIST | deep | linear | raw |
| CIFAR-10 | deep | linear | raw |
| EMNIST | linear | linear | raw |
| CIFAR-100 | linear | linear | pre-trained |

### F.4 BASELINES

We consider two baselines in our experiments for linear and deep setup.

**Linear Model:** In this setup, as we are focused on truncated IPS estimator, therefore we choose the Bayesian-CRM (B-CRM) method based on (London & Sandler, 2019, Proposition 1) introduced in (72). For B-CRM as our baseline, we estimate $\mu_0$ using logged known-reward dataset.

**Deep Model:** In this setup, we consider the BanditNet Joachims et al. (2018) as baseline. Note that, in BanditNet, instead of an IPS estimator, we have a self-normalized IPS (SNIPS) estimator. In particular, the SNIPS estimator is defined as

$$\text{SNIPS} := \frac{\sum_{i=1}^{n} r_i \frac{\pi_\theta(a_i|x_i)}{\pi_0(a_i|x_i)}}{\sum_{i=1}^{n} \frac{\pi_\theta(a_i|x_i)}{\pi_0(a_i|x_i)}}. \tag{74}$$

However, the SNIPS estimator in (74) can not be optimized by SGD and Joachims et al. (2018) proposed BanditNet as a constraint optimization version of (74) which can be optimized by SGD.

## F.5 RESULTS

For CIFAR-10 and FMNIST, in Tables 6 and 8, we compare the performance of all proposed algorithms, WCE-S2BL, KL-S2BL, WCE-S2BLK and KL-S2BLK in both deep and linear models with the baselines, BanditNet in deep model and Bayesian CRM in linear model for $\tau \in \{1, 5, 10, 20\}$ and $\rho \in \{0.02, 0.1, 0.2, 0.5, 1\}$. Similarly, for CIFAR-100 and EMNIST the results are presented in Tables 9 and 7. Note that, for $\rho = 1$, where we have access to all logged known-reward dataset, WCE-S2BL and KL-S2BL are the same as WCE-S2BLK and KL-S2BLK, respectively.

Table 6: Comparison of different algorithms WCE-S2BL, KL-S2BL, WCE-S2BLK, KL-S2BLK and BanditNet deterministic policy accuracy for FMNIST and CIFAR-10 with deep model setup and different qualities of logging policy ($\tau \in \{1, 5, 10, 20\}$) and proportions of labeled data ($\rho \in \{0.02, 0.1, 0.2, 0.5, 1\}$).

| Dataset | $\tau$ | $\rho$ | WCE-S2BL | KL-S2BL | WCE-S2BLK | KL-S2BLK | BanditNet | Logging Policy |
|---|---|---|---|---|---|---|---|---|
| FMNIST | 1 | 0.02 | **93.12 ± 0.16** | 91.79 ± 0.16 | 78.66 ± 0.90 | 61.46 ± 9.97 | 78.64 ± 1.97 | |
| | | 0.1 | **93.26 ± 0.05** | 91.73 ± 0.08 | 85.83 ± 0.85 | 77.75 ± 9.10 | 84.64 ± 4.24 | |
| | | 0.2 | **93.16 ± 0.18** | 92.04 ± 0.13 | 82.76 ± 4.45 | 87.72 ± 0.53 | 89.60 ± 0.49 | 91.73 |
| | | 0.5 | **93.19 ± 0.21** | 91.94 ± 0.04 | 88.72 ± 0.37 | 86.30 ± 1.43 | 91.59 ± 0.03 | |
| | | 1 | 93.10 ± 0.15 | 92.48 ± 0.6 | – | – | **93.54 ± 0.03** | |
| | 5 | 0.02 | **90.99 ± 0.09** | 83.54 ± 0.66 | 81.67 ± 0.36 | 34.27 ± 27.64 | 47.11 ± 12.51 | |
| | | 0.1 | **90.79 ± 0.14** | 81.65 ± 0.02 | 87.93 ± 0.07 | 73.48 ± 13.26 | 86.73 ± 0.63 | |
| | | 0.2 | **91.43 ± 0.07** | 82.71 ± 0.59 | 89.47 ± 0.06 | 88.94 ± 0.34 | 89.17 ± 0.26 | 53.97 |
| | | 0.5 | **91.74 ± 0.04** | 88.36 ± 0.15 | 89.18 ± 0.47 | 90.45 ± 0.12 | 90.42 ± 0.56 | |
| | | 1 | 91.41 ± 0.16 | 92.42 ± 0.12 | – | – | **92.65 ± 0.04** | |
| | 10 | 0.02 | **89.35 ± 0.15** | 69.94 ± 0.60 | 77.82 ± 0.73 | 45.18 ± 19.82 | 23.52 ± 3.15 | |
| | | 0.1 | **89.31 ± 0.16** | 80.68 ± 0.46 | 85.55 ± 0.39 | 80.54 ± 6.88 | 82.96 ± 3.03 | |
| | | 0.2 | **89.47 ± 0.3** | 79.45 ± 0.75 | 88.31 ± 0.14 | 67.53 ± 2.06 | 88.35 ± 0.45 | 20.72 |
| | | 0.5 | 90.05 ± 0.13 | 89.38 ± 0.13 | 89.81 ± 0.23 | 89.63 ± 0.98 | **90.44 ± 0.08** | |
| | | 1 | 91.00 ± 0.19 | 91.45 ± 0.17 | – | – | **92.21 ± 0.07** | |
| | 20 | 0.02 | **52.60 ± 1.36** | 45.20 ± 4.74 | 41.69 ± 2.19 | 22.23 ± 3.30 | 44.04 ± 7.50 | |
| | | 0.1 | **77.52 ± 0.63** | 76.89 ± 0.39 | 76.29 ± 0.48 | 71.28 ± 5.14 | 75.64 ± 1.65 | |
| | | 0.2 | **84.02 ± 0.28** | 82.25 ± 0.60 | 82.85 ± 0.94 | 82.69 ± 0.41 | 80.63 ± 0.13 | 10.54 |
| | | 0.5 | 86.83 ± 0.22 | 87.48 ± 0.26 | 86.51 ± 0.32 | 87.77 ± 0.17 | **87.61 ± 0.10** | |
| | | 1 | 87.05 ± 0.01 | 89.11 ± 0.10 | – | – | **89.03 ± 0.16** | |
| CIFAR-10 | 1 | 0.02 | **85.01 ± 0.37** | 84.6 ± 0.65 | 17.12 ± 0.97 | 21.63 ± 1.44 | 27.39 ± 3.47 | |
| | | 0.1 | 83.03 ± 1.49 | 84.34 ± 0.11 | 51.84 ± 0.92 | 46.24 ± 0.41 | 52.78 ± 0.56 | |
| | | 0.2 | 85.06 ± 0.32 | 85.53 ± 0.56 | 58.04 ± 5.47 | 54.12 ± 0.51 | 67.96 ± 0.62 | 79.77 |
| | | 0.5 | **84.79 ± 0.4** | 84.5 ± 0.09 | 79.23 ± 0.30 | 78.74 ± 0.56 | 71.36 ± 1.91 | |
| | | 1 | 84.63 ± 0.38 | 84.25 ± 0.45 | – | – | **86.82 ± 0.87** | |
| | 5 | 0.02 | **73.57 ± 0.36** | 51.14 ± 2.25 | 17.12 ± 0.97 | 21.63 ± 1.44 | 15.81 ± 5.12 | |
| | | 0.1 | **74.13 ± 1.43** | 58.31 ± 0.52 | 54.75 ± 0.39 | 33.21 ± 0.88 | 24.68 ± 3.74 | |
| | | 0.2 | **76.96 ± 0.35** | 63.19 ± 0.51 | 62.98 ± 0.81 | 46.35 ± 0.10 | 30.03 ± 12.75 | 53.97 |
| | | 0.5 | **76.46 ± 0.7** | 64.24 ± 2.09 | 70.50 ± 0.86 | 55.92 ± 0.66 | 58.34 ± 8.69 | |
| | | 1 | **77.53 ± 1.19** | 69.53 ± 1.09 | – | – | 70.12 ± 6.89 | |
| | 10 | 0.02 | **65.67 ± 1.06** | 37.8 ± 0.85 | 32.61 ± 1.14 | 20.66 ± 5.74 | 13.78 ± 1.99 | |
| | | 0.1 | **69.81 ± 0.87** | 43.00 ± 0.73 | 51.15 ± 0.64 | 35.87 ± 1.11 | 21.19 ± 3.35 | |
| | | 0.2 | **69.4 ± 0.47** | 48.44 ± 0.26 | 55.38 ± 3.63 | 44.60 ± 0.19 | 50.38 ± 0.55 | 43.45 |
| | | 0.5 | **75.08 ± 0.18** | 64.39 ± 0.05 | 71.90 ± 0.14 | 16.19 ± 0.99 | 68.92 ± 0.68 | |
| | | 1 | 75.58 ± 0.29 | **79.82 ± 0.36** | – | – | 78.8 ± 0.53 | |
| | 20 | 0.02 | **26.24 ± 1.42** | 15.09 ± 1.5 | 16.46 ± 1.77 | 12.56 ± 2.01 | 13.25 ± 1.36 | |
| | | 0.1 | **33.61 ± 0.58** | 30.67 ± 1.35 | 27.38 ± 2.44 | 27.74 ± 8.23 | 21.12 ± 1.01 | |
| | | 0.2 | 34.49 ± 4.01 | **36.95 ± 0.77** | 32.91 ± 6.95 | 34.27 ± 2.55 | 32.69 ± 2.17 | 20.72 |
| | | 0.5 | 46.95 ± 0.89 | **50.12 ± 4.43** | 47.69 ± 0.63 | 41.45 ± 9.93 | 36.79 ± 2.78 | |
| | | 1 | 47.68 ± 3.03 | **64.34 ± 0.85** | – | – | 55.27 ± 3.39 | |

## F.6 CIFAR-10 WITH PRE-TRAINED FEATURES

The linear experiments for CIFAR-10 and FashionMNIST, Table 1, are trained based on a deep logging policy. Due to the fact that the complexity of logging policy as a deep model is more than a linear model, the linear CIFAR-10 model accuracy is worse than the logging policy. However, our algorithms, WCE-S2BL and KL-S2BL, outperform the baseline, B-CRM, Table 1. The reason behind this setting is that a simple linear model doesn't work well on the raw flattened image and pre-trained features inject unknown prior information into the input of the models. To observe the difference between these two settings, in the experiments for the CIFAR-10 dataset, we use a linear model for both the logging policy and the trained policy, using pre-trained features as image representation in Table 10.

Table 7: Comparison of different algorithms WCE-S2BL, KL-S2BL, WCE-S2BLK, KL-S2BLK and BanditNet deterministic policy accuracy for EMNIST and CIFAR-100 with deep model setup and different qualities of logging policy ($\tau \in \{10, 20\}$ for EMNIST and $\tau \in \{1, 5, 10\}$ for CIFAR-100) and proportions of labeled data ($\rho \in \{0.02, 0.1, 0.2, 0.5, 1\}$).

| Dataset | $\tau$ | $\rho$ | WCE-S2BL | KL-S2BL | WCE-S2BLK | KL-S2BLK | BanditNet | Logging Policy |
|---|---|---|---|---|---|---|---|---|
| EMNIST | 10 | 0.02 | **98.77 ± 0.06** | 93.76 ± 0.46 | 93.25 ± 0.65 | 67.46 ± 40.63 | 95.44 ± 0.12 | |
| | | 0.1 | **98.75 ± 0.01** | 98.14 ± 0.15 | 96.61 ± 0.25 | 98.60 ± 0.10 | 98.46 ± 0.01 | |
| | | 0.2 | 98.81 ± 0.02 | 98.49 ± 0.01 | 98.13 ± 0.06 | 98.66 ± 0.04 | **99.11 ± 0.01** | 51.26 |
| | | 0.5 | 99.16 ± 0.04 | 99.03 ± 0.00 | 99.17 ± 0.01 | 99.09 ± 0.02 | **99.25 ± 0.07** | |
| | | 1 | 99.38 ± 0.05 | 99.39 ± 0.02 | – | – | **99.46 ± 0.02** | |
| | 20 | 0.02 | **96.54 ± 0.06** | 84.98 ± 3.04 | 79.49 ± 1.87 | 93.47 ± 0.50 | 89.50 ± 5.04 | |
| | | 0.1 | 97.79 ± 0.14 | 97.83 ± 0.02 | 97.88 ± 0.11 | **98.31.0.04** | 98.14 ± 0.12 | |
| | | 0.2 | 98.49 ± 0.04 | 98.33 ± 0.02 | 98.50 ± 0.05 | 98.42 ± 0.07 | **98.59 ± 0.05** | 25.58 |
| | | 0.5 | 98.83 ± 0.08 | 98.81 ± 0.03 | 98.79 ± 0.05 | 98.83 ± 0.11 | **99.07 ± 0.04** | |
| | | 1 | 99.08 ± 0.03 | **99.35 ± 0.03** | – | – | 99.16 ± 0.02 | |
| CIFAR-100 | 1 | 0.02 | **38.60 ± 0.23** | 15.67 ± 2.06 | 5.58 ± 1.06 | 1.76 ± 0.54 | 1.40 ± 0.29 | |
| | | 0.1 | **39.17 ± 0.65** | 17.02 ± 1.20 | 17.04 ± 1.50 | 16.21 ± 0.52 | 1.48 ± 0.23 | |
| | | 0.2 | **41.02 ± 0.54** | 18.36 ± 0.56 | 11.96 ± 0.45 | 22.18 ± 0.31 | 1.28 ± 0.21 | 26.48 |
| | | 0.5 | **41.42 ± 3.71** | 39.79 ± 0.24 | 32.13 ± 3.68 | 26.56 ± 3.46 | 1.91 ± 0.49 | |
| | | 1 | **42.93 ± 0.40** | 35.59 ± 1.79 | | | 2.09 ± 0.28 | |
| | 5 | 0.02 | **11.61 ± 0.83** | 3.04 ± 1.07 | 1.51 ± 0.50 | 1.0 ± 0.00 | 1.14 ± 0.20 | |
| | | 0.1 | **18.73 ± 0.78** | 4.76 ± 0.35 | 5.13 ± 0.15 | 1.46 ± 0.29 | 1.19 ± 0.27 | |
| | | 0.2 | **17.71 ± 0.07** | 4.30 ± 1.15 | 9.71 ± 0.79 | 1.26 ± 0.38 | 1.45 ± 0.12 | 4.58 |
| | | 0.5 | **19.13 ± 0.44** | 4.04 ± 0.17 | 15.64 ± 0.49 | 2.34 ± 0.30 | 1.32 ± 0,29 | |
| | | 1 | **19.71 ± 0.08** | 3.27 ± 1.15 | – | – | 1.5 ± 0.26 | |
| | 10 | 0.02 | **6.57 ± 0.64** | 1.50 ± 0.25 | 1.25 ± 0.18 | 1.0 ± 0.00 | 1.04 ± 0.06 | |
| | | 0.1 | **5.59 ± 0.48** | 1.75 ± 0.28 | 1.97 ± 0.27 | 1.22 ± 0.13 | 1.26 ± 0.15 | |
| | | 0.2 | **8.9 ± 0.32** | 1.86 ± 0.26 | 3.05 ± 0.10 | 1.52 ± 0.16 | 1.48 ± 0.24 | 1.73 |
| | | 0.5 | **8.21 ± 0.12** | 1.96 ± 0.24 | 6.28 ± 0.52 | 1.39 ± 0.21 | 1.36 ± 0.17 | |
| | | 1 | **9.22 ± 0.21** | 1.85 ± 0.23 | | | 1.39 ± 0.30 | |

## F.7 Real-World Experiments

We also carried out experiments on KuaiRec which is a dataset of human interactions with played videos in a mobile application. We adopt the setting introduced in Zhang et al. (2023a) for our experiments. Our logging policy is a random sampler choosing between items available for each user with random probabilities with the constraint to achieve $70\%$ average reward. We assign random scores in $[0.001, 1.0]$ to each item that the user rated and normalize items with the same reward together and multiply the score of items with reward 1 by $0.7$ and other items by $0.3$ to get the $70\%$ average reward. We don't use explicit truncation for propensity scores in this dataset. For each user, we sample 5 items according to the logging policy to create the logged bandit dataset.

Because in KuaiRec, as a recommendation system dataset, each user (context) can have multiple preferred items (actions), the accuracy of the learned policy (proportion of correctly suggested items) can't give a complete evaluation of the model's performance. We use the empirical IPS, evaluated based on test dataset.

We train the models with batch-size 32 and an initial learning rate of 0.01 with a cosine annealing learning rate scheduler and use automatic hyper-parameter tuning for other hyper-parameters. We repeat each experiment 5 times and report the average and standard deviation of scores. Table 11 shows our results.

## F.8 Discussion

The boost in performance is reliant on the quality of the initial logging policy. For example, both Tables 6 and 8 demonstrate that when the logging policy is nearly uniform (i.e., Large $\tau$), superior performance is predominantly realized through WCE-S2BL and KL-S2BL. It should be noted that the improvement is also dependent on the available portion of the logged known-reward dataset, denoted as $\rho$. For example, it is observed that in the majority of cases, when we have access to a relatively minor segment of the logged known-reward dataset (e.g., $\rho = 0.02$), the performance of WCE-S2BL is superior. This superior performance is particularly evident within the FMNIST and CIFAR-10 dataset for deep model, where WCE-S2BL typically surpasses the performance of other proposed methods and B-CRM as baseline. In the Linear model at $\tau = 1$, wherein the performance of the logging policy exceeds $90\%$, there is an absence of algorithms demonstrating superior performance relative to the logging policy. It can be due to complexity of feature space and the limitation of linear

Table 8: Comparison of different algorithms WCE-S2BL, KL-S2BL, WCE-S2BLK, KL-S2BLK and Bayesian-CRM (B-CRM) deterministic policy accuracy for FMNIST and CIFAR-10 with linear model setup and different qualities of logging policy ($\tau \in \{1, 5, 10, 20\}$) and proportions of labeled data ($\rho \in \{0.02, 0.1, 0.2, 0.5, 1\}$).

| Dataset | $\tau$ | $\rho$ | WCE-S2BL | KL-S2BL | WCE-S2BLK | KL-S2BLK | B-CRM | Logging Policy |
|---|---|---|---|---|---|---|---|---|
| FMNIST | 1 | 0.02 | $84.37 \pm 0.14$ | $71.67 \pm 0.26$ | $78.84 \pm 0.05$ | $74.71 \pm 0.06$ | $64.67 \pm 1.44$ | |
| | | 0.1 | $84.18 \pm 0.00$ | $75.43 \pm 0.04$ | $82.35 \pm 0.05$ | $72.45 \pm 0.01$ | $70.38 \pm 0.09$ | |
| | | 0.2 | $83.59 \pm 0.18$ | $71.88 \pm 0.31$ | $83.05 \pm 0.06$ | $74.06 \pm 0.00$ | $70.99 \pm 0.32$ | **91.73** |
| | | 0.5 | $84.14 \pm 0.20$ | $71.03 \pm 0.13$ | $83.85 \pm 0.00$ | $71.05 \pm 1.79$ | $71.76 \pm 0.03$ | |
| | | 1 | $84.24 \pm 0.07$ | $69.44 \pm 1.20$ | $-$ | $-$ | $72.42 \pm 0.01$ | |
| | 5 | 0.02 | $\mathbf{83.51 \pm 0.01}$ | $19.60 \pm 0.42$ | $75.24 \pm 2.89$ | $19.48 \pm 0.33$ | $64.49 \pm 01.04$ | |
| | | 0.1 | $\mathbf{83.99 \pm 0.02}$ | $36.33 \pm 11.60$ | $80.11 \pm 0.09$ | $29.55 \pm 3.72$ | $70.21 \pm 0.07$ | |
| | | 0.2 | $\mathbf{83.91 \pm 0.07}$ | $54.83 \pm 1.68$ | $82.69 \pm 0.19$ | $51.02 \pm 9.33$ | $71.14 \pm 0.10$ | 53.97 |
| | | 0.5 | $\mathbf{83.91 \pm 0.01}$ | $59.49 \pm 0.61$ | $83.47 \pm 0.02$ | $72.13 \pm 0.24$ | $71.86 \pm 0.14$ | |
| | | 1 | $\mathbf{83.62 \pm 0.01}$ | $73.11 \pm 0.60$ | $-$ | $-$ | $72.33 \pm 0.06$ | |
| | 10 | 0.02 | $\mathbf{82.31 \pm 0.07}$ | $26.71 \pm 2.18$ | $77.43 \pm 0.13$ | $18.35 \pm 7.06$ | $66.24 \pm 00.03$ | |
| | | 0.1 | $\mathbf{82.30 \pm 0.04}$ | $56.51 \pm 9.65$ | $77.59 \pm 0.34$ | $47.93 \pm 5.15$ | $70.33 \pm 0.33$ | |
| | | 0.2 | $\mathbf{83.15 \pm 0.09}$ | $67.10 \pm 5.17$ | $81.20 \pm 0.12$ | $60.26 \pm 0.88$ | $71.02 \pm 0.30$ | 20.72 |
| | | 0.5 | $\mathbf{83.27 \pm 0.01}$ | $74.97 \pm 0.17$ | $82.85 \pm 0.10$ | $70.02 \pm 1.41$ | $71.72 \pm 0.01$ | |
| | | 1 | $\mathbf{83.00 \pm 0.09}$ | $73.92 \pm 0.27$ | $-$ | $-$ | $72.25 \pm 0.10$ | |
| | 20 | 0.02 | $47.44 \pm 2.83$ | $32.24 \pm 4.95$ | $51.21 \pm 2.52$ | $21.66 \pm 1.89$ | $\mathbf{63.99 \pm 1.01}$ | |
| | | 0.1 | $\mathbf{75.10 \pm 0.09}$ | $69.22 \pm 4.09$ | $75.02 \pm 0.04$ | $59.04 \pm 0.59$ | $68.43 \pm 0.33$ | |
| | | 0.2 | $77.19 \pm 0.02$ | $74.43 \pm 0.78$ | $\mathbf{77.36 \pm 0.02}$ | $73.36 \pm 1.51$ | $69.21 \pm 0.24$ | 10.54 |
| | | 0.5 | $73.89 \pm 0.00$ | $\mathbf{79.04 \pm 0.17}$ | $77.5 \pm 0.17$ | $78.92 \pm 0.04$ | $71.17 \pm 0.05$ | |
| | | 1 | $\mathbf{78.51 \pm 0.01}$ | $74.36 \pm 0.01$ | $-$ | $-$ | $71.74 \pm 0.16$ | |
| CIFAR-10 | 1 | 0.02 | $39.39 \pm 0.15$ | $37.21 \pm 0.15$ | $30.56 \pm 0.61$ | $30.08 \pm 0.27$ | $19.00 \pm 1.77$ | |
| | | 0.1 | $40.18 \pm 0.08$ | $37.74 \pm 0.02$ | $35.76 \pm 0.04$ | $33.42 \pm 0.24$ | $27.72 \pm 0.37$ | |
| | | 0.2 | $40.66 \pm 0.29$ | $37.88 \pm 0.58$ | $38.22 \pm 0.01$ | $35.70 \pm 0.25$ | $29.32 \pm 0.35$ | **79.77** |
| | | 0.5 | $40.81 \pm 0.08$ | $38.55 \pm 0.14$ | $39.64 \pm 0.14$ | $36.97 \pm 0.06$ | $30.67 \pm 0.28$ | |
| | | 1 | $40.77 \pm 0.01$ | $38.07 \pm 0.42$ | $-$ | $-$ | $31.32 \pm 0.36$ | |
| | 5 | 0.02 | $34.60 \pm 0.06$ | $10.26 \pm 0.37$ | $14.18 \pm 5.92$ | $10.00 \pm 0.00$ | $12.76 \pm 3.07$ | |
| | | 0.1 | $39.91 \pm 0.84$ | $10.90 \pm 1.02$ | $35.08 \pm 0.08$ | $10.40 \pm 0.57$ | $24.50 \pm 1.00$ | |
| | | 0.2 | $40.15 \pm 0.06$ | $11.58 \pm 2.09$ | $37.50 \pm 1.09$ | $11.52 \pm 2.15$ | $27.70 \pm 0.47$ | **53.97** |
| | | 0.5 | $39.90 \pm 0.54$ | $31.61 \pm 0.19$ | $38.67 \pm 0.03$ | $20.51 \pm 0.17$ | $29.50 \pm 0.19$ | |
| | | 1 | $40.52 \pm 0.07$ | $32.50 \pm 0.84$ | $-$ | $-$ | $30.22 \pm 0.81$ | |
| | 10 | 0.02 | $38.97 \pm 0.03$ | $10.84 \pm 1.18$ | $26.60 \pm 0.89$ | $10.03 \pm 0.05$ | $14.17 \pm 2.87$ | |
| | | 0.1 | $39.04 \pm 0.02$ | $14.70 \pm 5.24$ | $34.42 \pm 0.10$ | $11.54 \pm 2.18$ | $24.17 \pm 3.25$ | |
| | | 0.2 | $39.69 \pm 0.05$ | $15.49 \pm 2.23$ | $35.71 \pm 0.73$ | $13.81 \pm 2.74$ | $28.24 \pm 0.20$ | **43.45** |
| | | 0.5 | $39.57 \pm 0.12$ | $28.52 \pm 0.36$ | $38.53 \pm 0.36$ | $20.80 \pm 0.28$ | $29.78 \pm 0.42$ | |
| | | 1 | $38.87 \pm 0.23$ | $28.07 \pm 0.92$ | $-$ | $-$ | $30.09 \pm 0.47$ | |
| | 20 | 0.02 | $17.03 \pm 0.08$ | $11.1 \pm 1.56$ | $16.39 \pm 0.68$ | $10.01 \pm 0.02$ | $10.25 \pm 0.07$ | |
| | | 0.1 | $20.46 \pm 0.03$ | $11.54 \pm 1.59$ | $18.87 \pm 0.04$ | $11.46 \pm 1.47$ | $15.28 \pm 1.963$ | |
| | | 0.2 | $\mathbf{22.06 \pm 0.17}$ | $10.55 \pm 0.65$ | $20.23 \pm 0.02$ | $12.92 \pm 1.03$ | $19.58 \pm 1.10$ | 20.72 |
| | | 0.5 | $23.30 \pm 0.14$ | $14.35 \pm 0.59$ | $21.11 \pm 0.16$ | $16.99 \pm 0.79$ | $\mathbf{24.77 \pm 1.69}$ | |
| | | 1 | $25.35 \pm 0.12$ | $23.21 \pm 0.19$ | $-$ | $-$ | $\mathbf{25.40 \pm 2.03}$ | |

model. The same phenomena is also observed in CIFAR-10 for linear model. In deep model setup, we observe that WCE-S2BL for $\tau = 1$ and FMNIST has better performance with respect to other proposed method.

It is worthwhile to mention that for the logging policy close to uniform, our methods have better performance in both linear and deep models.

Regarding the performance of WCE-S2BLK and KL-S2BLK with respect to WCE-S2BL and KL-S2BL, we can observe that in all cases, the logged missing-reward dataset, can help us to achieve a better performance. From Proposition 5, we expected that the error of estimators of KL divergence and reverse KL divergence would reduce by using more data samples. Therefore, the logged missing-reward dataset, can help to minimize the KL divergence and reverse KL divergence with a better estimation error.

Given that the same logged data were applied in both the linear and deep models, an apparent observation is the enhanced performance displayed by the deep model for all datasets, i.e., FMNIST, CIFAR-10, CIFAR-100 and EMNIST.

Table 9: Comparison of different algorithms WCE-S2BL, KL-S2BL, WCE-S2BLK, KL-S2BLK and Bayesian-CRM (B-CRM) deterministic policy accuracy for EMNIST and CIFAR-100 with linear model setup and different qualities of logging policy ($\tau \in \{1, 5, 10, 20\}$) and proportions of labeled data ($\rho \in \{0.02, 0.1, 0.2, 0.5, 1\}$).

| Dataset | $\tau$ | $\rho$ | WCE-S2BL | KL-S2BL | WCE-S2BLK | KL-S2BLK | B-CRM | Logging Policy |
|---------|------|------|----------|---------|-----------|----------|-------|----------------|
| EMNIST | 1 | 0.02 | **87.00 ± 0.01** | 77.18 ± 0.37 | 86.10 ± 0.06 | 52.52 ± 0.68 | 76.91 ± 0.12 | |
| | | 0.1 | **87.52 ± 0.00** | 69.79 ± 0.56 | 86.92 ± 0.07 | 52.80 ± 1.65 | 80.84 ± 0.07 | |
| | | 0.2 | **87.60 ± 0.01** | 79.83 ± 0.50 | 87.46 ± 0.04 | 76.11 ± 0.69 | 81.61 ± 0.08 | 76.55 |
| | | 0.5 | 87.69 ± 0.04 | 76.52 ± 0.42 | **87.71 ± 0.03** | 77.79 ± 0.30 | 82.02 ± 0.09 | |
| | | 1 | **87.68 ± 0.02** | 80.83 ± 0.73 | – | – | 82.57 ± 0.01 | |
| | 5 | 0.02 | **74.14 ± 0.02** | 33.86 ± 0.38 | 70.68 ± 0.03 | 15.14 ± 4.23 | 56.13 ± 0.42 | |
| | | 0.1 | **82.10 ± 2.21** | 59.92 ± 0.57 | 62.42 ± 0.33 | 49.00 ± 1.58 | 62.79 ± 0.20 | |
| | | 0.2 | **82.21 ± 2.60** | 69.39 ± 0.37 | 77.55 ± 4.55 | 51.28 ± 6.94 | 68.21 ± 0.22 | 41.06 |
| | | 0.5 | **84.91 ± 2.87** | 85.22 ± 0.13 | 76.38 ± 3.00 | 68.51 ± 6.12 | 73.12 ± 0.17 | |
| | | 1 | 80.03 ± 2.03 | **86.81 ± 0.05** | – | – | 75.38 ± 0.25 | |
| | 10 | 0.02 | **82.91 ± 0.01** | 33.54 ± 1.24 | 80.08 ± 0.04 | 9.67 ± 0.38 | 55.89 ± 0.05 | |
| | | 0.1 | **82.95 ± 0.03** | 55.02 ± 0.79 | 82.40 ± 0.01 | 35.56 ± 0.97 | 65.94 ± 0.33 | |
| | | 0.2 | 83.9 ± 3.19 | **84.27 ± 0.07** | 79.15 ± 0.04 | 83.16 ± 0.31 | 69.70 ± 0.17 | 31.86 |
| | | 0.5 | **88.01 ± 0.15** | 86.42 ± 0.04 | 85.28 ± 2.68 | 86.40 ± 0.02 | 73.43 ± 0.20 | |
| | | 1 | **88.98 ± 0.35** | 86.77 ± 0.01 | – | – | 75.18 ± 0.19 | |
| | 20 | 0.02 | **82.17 ± 0.04** | 23.34 ± 0.40 | 78.25 ± 0.16 | 22.71 ± 2.07 | 54.02 ± 0.93 | |
| | | 0.1 | **87.72 ± 0.14** | 63.02 ± 2.19 | 86.89 ± 0.01 | 56.22 ± 2.29 | 67.20 ± 0.46 | |
| | | 0.2 | **88.66 ± 0.06** | 82.93 ± 0.25 | 84.06 ± 0.05 | 82.21 ± 0.32 | 70.70 ± 0.10 | 23.83 |
| | | 0.5 | 89.66 ± 0.09 | 84.76 ± 0.14 | **89.78 ± 0.05** | 84.18 ± 0.03 | 73.94 ± 0.12 | |
| | | 1 | **89.37 ± 0.17** | 80.00 ± 0.10 | – | – | 76.08 ± 0.05 | |
| CIFAR-100 | 1 | 0.02 | **13.59 ± 0.08** | 6.81 ± 2.94 | 11.92 ± 0.24 | 2.59 ± 2.24 | 4.23 ± 0.26 | |
| | | 0.1 | **13.65 ± 0.05** | 7.53 ± 0.35 | 12.60 ± 0.06 | 4.73 ± 0.51 | 8.29 ± 0.05 | |
| | | 0.2 | **13.73 ± 0.02** | 9.07 ± 0.53 | 13.48 ± 0.02 | 5.73 ± 0.27 | 9.45 ± 0.13 | 12.32 |
| | | 0.5 | **13.70 ± 0.02** | 11.46 ± 0.33 | 13.56 ± 0.10 | 9.89 ± 0.43 | 10.94 ± 0.03 | |
| | | 1 | **13.75 ± 0.06** | 11.99 ± 0.99 | – | – | 12.32 ± 0.03 | |
| | 5 | 0.02 | **16.38 ± 0.04** | 1.85 ± 1.21 | 2.18 ± 0.32 | 1.0 ± 0.00 | 3.56 ± 0.01 | |
| | | 0.1 | **16.14 ± 0.05** | 2.83 ± 1.39 | 15.30 ± 0.07 | 1.06 ± 0.08 | 6.68 ± 0.09 | |
| | | 0.2 | **16.63 ± 0.04** | 4.67 ± 0.70 | 16.13 ± 0.10 | 1.90 ± 0.60 | 8.46 ± 0.23 | 6.01 |
| | | 0.5 | **16.62 ± 0.04** | 9.57 ± 0.48 | 16.49 ± 0.06 | 4.90 ± 0.34 | 9.83 ± 0.09 | |
| | | 1 | **16.90 ± 0.01** | 12.24 ± 0.29 | – | – | 10.86 ± 0.07 | |
| | 10 | 0.02 | **15.43 ± 0.30** | 1.25 ± 0.36 | 1.17 ± 0.27 | 1.0 ± 0.00 | 2.99 ± 0.15 | |
| | | 0.1 | **15.90 ± 0.11** | 2.47 ± 0.67 | 10.81 ± 0.03 | 0.99 ± 0.01 | 6.01 ± 0.11 | |
| | | 0.2 | **16.21 ± 0.09** | 2.84 ± 0.18 | 14.53 ± 0.08 | 1.44 ± 0.20 | 7.62 ± 0.09 | 3.4 |
| | | 0.5 | **16.71 ± 0.06** | 6.17 ± 0.60 | 16.80 ± 0.04 | 3.10 ± 0.56 | 8.77 ± 0.06 | |
| | | 1 | **16.87 ± 0.08** | 6.04 ± 1.18 | – | – | 10.25 ± 0.09 | |
| | 20 | 0.02 | **18.65 ± 0.06** | 1.00 ± 0.00 | 1.36 ± 0.27 | 1.17 ± 0.16 | 4.16 ± 0.14 | |
| | | 0.1 | **18.29 ± 0.01** | 1.60 ± 0.42 | 12.43 ± 0.09 | 1.07 ± 0.07 | 7.46 ± 0.05 | |
| | | 0.2 | **16.99 ± 0.10** | 2.90 ± 0.34 | 5.31 ± 0.23 | 1.06 ± 0.09 | 8.80 ± 0.09 | 3.22 |
| | | 0.5 | **19.43 ± 0.02** | 4.38 ± 0.21 | 19.22 ± 0.05 | 2.19 ± 0.70 | 10.23 ± 0.15 | |
| | | 1 | **20.36 ± 0.11** | 5.96 ± 1.01 | – | – | 11.43 ± 0.01 | |

## F.9 EFFECT OF LOGGED MISSING-REWARD DATASET AND MINIMIZATION OF THE REGULARIZATION

We also run experiments to investigate the effect of logged missing-reward dataset size. For this purpose, we fix the number of logged known-reward dataset to 1000 samples for CIFAR-10 and 1200 for FMNIST. Then, we add 1000, 4000, 9000, 24000 missing-reward samples to the dataset and compute the accuracy of the parameterized learned policy. Figure 3 shows the accuracy for different numbers of added missed-reward samples for CIFAR-10 and FMNIST datasets over different ratio of logged missing-reward samples to logged known-reward samples. We can observe that by increasing the number of missing-reward logged data samples (the ratio of logged missing-reward samples to logged known-reward samples with fixed logged known-reward sample size), the deterministic accuracy is improved. To provide more insight with respect to minimization of regularization, we run some experiments for deep model, to investigate the performance if we just minimize the regularization terms, i.e., KL divergence or reverse KL divergence, via the logged known-reward dataset and missing-reward datasets. The results are shown in Table 12. It can be noted that, under all circumstances, it is essential to minimize the regularized version of BanditNet for better accuracy. Therefore, both main loss and regularization are needed for better performance.

Table 10: Comparison of different algorithms WCE-S2BL, KL-S2BL, WCE-S2BLK, KL-S2BLK and Bayesian-CRM (B-CRM) deterministic policy accuracy for CIFAR-10 with linear model setup and different qualities of logging policy ($\tau \in \{1, 5, 10, 20\}$) and proportions of labeled data ($\rho \in \{0.02, 0.1, 0.2, 0.5, 1\}$).

| Dataset | $\tau$ | $\rho$ | WCE-S2BL | KL-S2BL | WCE-S2BLK | KL-S2BLK | B-CRM | Logging Policy |
|---|---|---|---|---|---|---|---|---|
| CIFAR-10 | 1 | 0.02 | **62.95 ± 0.08** | 28.29 ± 11.35 | 9.49 ± 0.72 | 10.02 ± 0.02 | 55.02 ± 0.14 | |
| | | 0.1 | **62.97 ± 0.27** | 21.79 ± 14.50 | 60.89 ± 0.13 | 10.00 ± 0.00 | 56.59 ± 0.26 | |
| | | 0.2 | **62.83 ± 0.06** | 26.29 ± 6.92 | 62.90 ± 0.10 | 14.08 ± 1.58 | 57.75 ± 0.42 | 52.89 |
| | | 0.5 | **63.49 ± 0.07** | 46.15 ± 1.16 | 62.89 ± 0.07 | 43.25 ± 1.21 | 58.81 ± 0.03 | |
| | | 1 | **63.85 ± 0.09** | 44.07 ± 0.79 | − | − | 59.24 ± 0.05 | |
| | 5 | 0.02 | **56.84 ± 0.07** | 15.03 ± 6.04 | 51.40 ± 0.29 | 13.74 ± 2.47 | 44.48 ± 1.02 | |
| | | 0.1 | **57.47 ± 0.28** | 17.69 ± 1.21 | 55.85 ± 0.19 | 9.96 ± 0.06 | 50.88 ± 0.32 | |
| | | 0.2 | **58.50 ± 0.02** | 18.74 ± 1.58 | 58.22 ± 0.05 | 13.26 ± 3.54 | 52.56 ± 0.21 | 40.96 |
| | | 0.5 | **59.47 ± 0.09** | 23.60 ± 2.68 | 60.03 ± 0.03 | 18.06 ± 2.84 | 53.44 ± 0.28 | |
| | | 1 | **60.97 ± 0.01** | 34.35 ± 2.59 | − | − | 54.13 ± 0.09 | |
| | 10 | 0.02 | **54.47 ± 1.34** | 11.60 ± 1.13 | 41.93 ± 1.25 | 10.08 ± 0.12 | 44.66 ± 0.29 | |
| | | 0.1 | **55.47 ± 0.29** | 19.19 ± 0.19 | 54.84 ± 0.02 | 10.00 ± 0.00 | 50.76 ± 0.25 | |
| | | 0.2 | **56.99 ± 0.00** | 22.83 ± 0.46 | 56.94 ± 0.19 | 13.69 ± 2.69 | 52.09 ± 0.43 | 36.6 |
| | | 0.5 | 60.27 ± 0.08 | 30.11 ± 3.22 | **60.77 ± 0.00** | 24.60 ± 2.35 | 53.19 ± 0.42 | |
| | | 1 | **61.14 ± 0.04** | 40.54 ± 0.48 | − | − | 53.75 ± 0.14 | |
| | 20 | 0.02 | **56.33 ± 0.16** | 13.92 ± 5.55 | 46.27 ± 2.51 | 10.00 ± 0.00 | 45.11 ± 0.82 | |
| | | 0.1 | **57.23 ± 0.00** | 20.79 ± 0.03 | 56.43 ± 0.18 | 13.92 ± 0.52 | 50.69 ± 0.43 | |
| | | 0.2 | 57.87 ± 0.11 | 16.3 ± 4.20 | **57.90 ± 0.27** | 11.73 ± 1.90 | 51.88 ± 0.22 | 41.63 |
| | | 0.5 | 59.05 ± 0.14 | 24.16 ± 0.67 | **59.10 ± 0.30** | 19.23 ± 0.37 | 53.08 ± 0.14 | |
| | | 1 | **61.76 ± 0.16** | 33.98 ± 0.88 | − | − | 53.51 ± 0.16 | |

Table 11: Comparison of different algorithms WCE-S2BL, KL-S2BL, BanditNet empirical IPS for KuaiRec with different proportions of labeled data ($\rho \in \{0.02, 0.1, 0.2, 0.5, 1\}$).

| Dataset | $\rho$ | WCE-S2BL | KL-S2BL | BanditNet |
|---|---|---|---|---|
| KuaiRec | 0.02 | 0.73 ± 0.33 | **0.88 ± 0.28** | 0.74 ± 0.43 |
| | 0.1 | **0.73 ± 0.27** | 0.69 ± 0.19 | 0.58 ± 0.09 |
| | 0.2 | **0.70 ± 0.13** | 0.62 ± 0.19 | 0.69 ± 0.34 |
| | 0.5 | **0.76 ± 0.26** | 0.72 ± 0.20 | 0.66 ± 0.17 |
| | 1.0 | **0.94 ± 0.20** | 0.73 ± 0.12 | 0.66 ± 0.23 |

## F.10  CODE

We thank the authors of Aouali et al. (2023) for kindly sharing their code with us. The code for this study is written in Python. We use Pytorch for the training of our model. The supplementary material includes a zip file named rl_without_reward.zip with the following files:

- **preprocess_raw_dataset_from_model.py**: The code to generate the base pre-processed version of the datasets with raw input values.

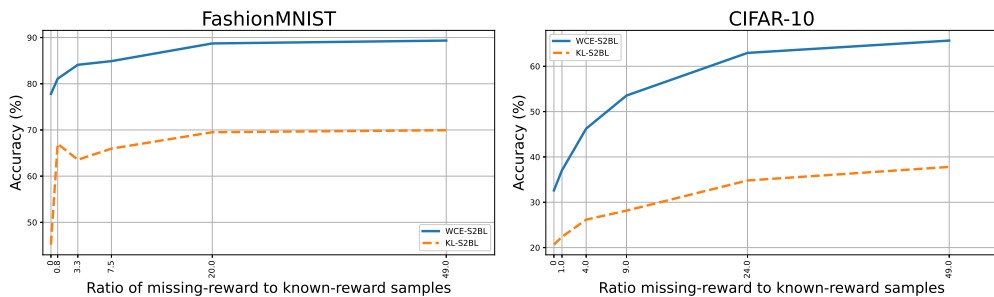

Figure 3: Accuracy of WCE-S2BL and KL-S2BL for different ratio of missing-reward data samples to known-reward data samples. We fix the number of known-reward data samples to 1000 samples.

Table 12: Comparison of WCE-S2BL, KL-S2BL deterministic accuracy trained with $\rho = 0.1$ and their counterpart without Self- normalized IPS (SNIPS) as main loss in BanditNet. Accuracy on FMNIST and CIFAR-10 datasets is reported for $\tau \in \{1, 5, 10, 20\}$.

| Dataset | $\tau$ | WCE-S2BL | WCE-S2BL w/o SNIPS | KL-S2BL | KL-S2BL w/o SNIPS |
|---|---|---|---|---|---|
| FMNIST | 1 | $93.26 \pm 0.05$ | $92.89 \pm 0.09$ | $91.73 \pm 0.08$ | $85.51 \pm 1.20$ |
| | 5 | $90.79 \pm 0.14$ | $90.69 \pm 0.19$ | $81.65 \pm 0.02$ | $78.15 \pm 1.57$ |
| | 10 | $89.31 \pm 0.16$ | $88.32 \pm 0.16$ | $80.68 \pm 0.46$ | $74.76 \pm 0.73$ |
| | 20 | $77.52 \pm 0.63$ | $14.15 \pm 0.50$ | $76.89 \pm 0.39$ | $13.86 \pm 0.95$ |
| CIFAR-10 | 1 | $83.03 \pm 1.49$ | $83.34 \pm 0.09$ | $84.34 \pm 0.11$ | $64.69 \pm 0.6$ |
| | 5 | $74.13 \pm 1.43$ | $72.87 \pm 0.76$ | $58.31 \pm 0.52$ | $59.30 \pm 0.56$ |
| | 10 | $69.81 \pm 0.87$ | $66.75 \pm 0.91$ | $43.00 \pm 0.73$ | $42.92 \pm 0.44$ |
| | 20 | $33.61 \pm 0.58$ | $28.40 \pm 0.07$ | $30.67 \pm 1.35$ | $10.89 \pm 0.71$ |

- **preprocess_feature_dataset_from_model.py**: The code to generate the base pre-processed version of the datasets with pre-trained features.
- The **data** folder consists of any potentially generated bandit dataset (which can be generated by running the scripts in code).
- The **code** folder contains the scripts and codes written for the experiments.
  - **requirements.txt** contains the Python libraries required to reproduce our results.
  - **readme.md** includes the syntax of different commands in the code.
  - **accs**: A folder containing the result reports of different experiments.
  - **saved_logs**: Training log for different experiments.
  - **data.py & data_rec.py** code to load data for image datasets and KuaiRec dataset respectively.
  - **config**: Contains different configuration files for different setups.
  - **runs**: Folder containing different batch running scripts.
  - **train_logging_policy.py**: Script to train the logging policy.
  - **create_bandit_dataset.py**: Code for the generation of the bandit dataset using the logging policy.
  - **main_semi_ot.py**: Main training code which implements different methods proposed by our paper.
  - **main_semi_rec.py**: Main training code for KuaiRec dataset.
  - **create_kuairec_dataset.ipynb**: Code for preparing and preprocessing KuaiRec dataset.

To use this code, the user needs to first download the CIFAR-10 dataset from this Link and make sure that the data_batch files are inside the folder *data/cifar/*. Then, install the Python libraries listed in the file *requirements.txt*. Then run the preprocess scripts to convert the datasets to the appropriate format. For the FashionMNIST dataset, no manual download is required.

All our experiments were run using 3 servers, each one with a GTX 3090 GPU and 32GB of RAM,

## G   TRUE RISK REGULARIZATION

We can choose the KL divergence instead of the square root of the KL divergence as a regularizer for IPS estimator minimization. In this section, we study the true risk regularization using KL divergence $\mathrm{KL}(\pi_\theta(A|X)\|\pi_0(A|X))$, as follows:

$$\min_{\pi_\theta} R(\pi_\theta) + \lambda \mathrm{KL}(\pi_\theta(A|X)\|\pi_0(A|X)), \quad \lambda \geq 0. \tag{75}$$

It is possible to provide the optimal solution to regularized minimization (75).

**Theorem 2.** *Considering the true risk minimization with KL divergence regularization,*

$$\min_{\pi_\theta} R(\pi_\theta) + \lambda \mathrm{KL}(\pi_\theta(A|X)\|\pi_0(A|X)), \quad \lambda \geq 0, \tag{76}$$

*the optimal parameterized policy is:*

$$\pi_\theta^\star(A = a|X = x) = \frac{\pi_0(A = a|X = x)e^{-\frac{1}{\lambda}r(a,x)}}{\mathbb{E}_{\pi_0}[e^{-\frac{1}{\lambda}r(a,x)}]} \tag{77}$$

*Proof.* The minimization problem (75) can be written as follows:

$$\min_{\pi_\theta} \mathbb{E}_{P_X}[\mathbb{E}_{\pi_\theta(A|X)}[r(A, X)]] + \lambda\mathrm{KL}(\pi_\theta(A|X)\|\pi_0(A|X)), \quad \lambda \geq 0 \tag{78}$$

Using the same approach by Zhang (2006); Aminian et al. (2021) and considering $\frac{1}{\lambda}$ as the inverse temperature, the final result holds. $\square$

The optimal parameterized policy under KL divergence regularization, i.e.,

$$\pi_\theta^\star(A = a|X = x) = \frac{\pi_0(A = a|X = x)e^{-\frac{1}{\lambda}r(a,x)}}{\mathbb{E}_{\pi_0}[e^{-\frac{1}{\lambda}r(a,x)}]} \tag{79}$$

provides the following insights:

- The optimal parameterized policy, $\pi_\theta^\star(A|X)$, is a stochastic policy.
- The optimal parameterized policy is invariant with respect to constant shifts in the reward function.
- For asymptotic condition, i.e., $\lambda \to 0$, the optimal parameterized policy will be deterministic policy.

## H  FURTHER DISCUSSION

### H.1  Q-LEARNING APPROACH

Inspired by Pseudo-labeling approach in semi-supervised learning, we can propose Q-learning approach (reward-function estimation). In this approach, we first estimate the reward function using a logged known-reward dataset. Using the model for reward function, we assign pseudo-rewards to the logged missing-reward dataset. Then we train the final model via truncated IPS estimator using both logged known-reward and Pseudo-reward datasets.

For experiments, we employed a logistic regression with a sigmoid activation function and a linear layer. Note that, in this scenario the rewards are binary. Second, we generate pseudo-rewards by applying the reward function estimator to the logged missing-reward dataset. Finally, we train the truncated IPS estimator with both the logged known-reward dataset and the pseudo-reward dataset.

In Table 13, we present the results (accuracy) of our algorithms (WCE-S2BL and KL-S2BL) and Q-learning under the EMNIST dataset with varying ratios of missing-reward data to known-reward data.

As we can observe, the performance of Q-learning approach in EMNIST is worse than our algorithms, WCE-S2BL and KL-S2BL. Note that the Pseudo-reward for logged missing-reward samples be different from true reward. Therefore, we have some noise in reward and the (truncated) IPS estimator underperforms under noisy-rewards, Wang et al. (2017). It is interesting to explore other estimator which are robust to noise in rewards and can improve the Q-learning approach under both known-reward and missing-reward datasets.

### H.2  LOGGED MISSING-REWARD DATASET AND REGULARIZATION ESTIMATORS

We conducted experiments on WCE-S2BL and KL-S2BL by utilizing the logged known-reward dataset for regularization, denoting these algorithms as WCE-S2BLK and KL-S2BLK, respectively, Appendix F. In these WCE-S2BLK and KL-S2BLK variations, our focus is directed towards minimizing the (estimated) KL divergence or reverse-KL divergence solely via the logged known-reward dataset.

Table 13: Comparison of different algorithms WCE-S2BL, KL-S2BL and Q-learning for EMNIST with linear model setup and different qualities of logging policy ($\tau \in \{1, 5, 10, 20\}$) and proportions of labeled data ($\rho \in \{0.02, 0.1, 0.2, 0.5, 1\}$).

| Dataset | $\tau$ | $\rho$ | WCE-S2BL | KL-S2BL | Q-Learning | Logging Policy |
|---|---|---|---|---|---|---|
| EMNIST | 1 | 0.02 | $\mathbf{87.00 \pm 0.01}$ | $77.18 \pm 0.37$ | $26.16 \pm 1.30$ | 76.55 |
| | | 0.1 | $\mathbf{87.52 \pm 0.00}$ | $69.79 \pm 0.56$ | $22.34 \pm 0.48$ | |
| | | 0.2 | $\mathbf{87.60 \pm 0.01}$ | $79.83 \pm 0.50$ | $21.99 \pm 0.93$ | |
| | | 0.5 | $\mathbf{87.69 \pm 0.04}$ | $76.52 \pm 0.42$ | $11.17 \pm 0.25$ | |
| | | 1.0 | $\mathbf{87.68 \pm 0.02}$ | $80.83 \pm 0.73$ | $10.00 \pm 0.00$ | |
| | 5 | 0.02 | $\mathbf{74.14 \pm 0.02}$ | $33.86 \pm 0.38$ | $10.0 \pm 0.00$ | 41.06 |
| | | 0.1 | $\mathbf{82.10 \pm 2.21}$ | $59.92 \pm 0.57$ | $21.37 \pm 4.35$ | |
| | | 0.2 | $\mathbf{82.21 \pm 2.60}$ | $69.39 \pm 0.37$ | $12.74 \pm 3.87$ | |
| | | 0.5 | $84.91 \pm 2.87$ | $\mathbf{85.22 \pm 0.13}$ | $59.80 \pm 5.12$ | |
| | | 1.0 | $80.03 \pm 2.03$ | $\mathbf{86.81 \pm 0.05}$ | $81.08 \pm 7.16$ | |
| | 10 | 0.02 | $\mathbf{82.91 \pm 0.01}$ | $33.54 \pm 1.24$ | $30.43 \pm 4.50$ | 31.86 |
| | | 0.1 | $\mathbf{82.95 \pm 0.03}$ | $55.02 \pm 0.79$ | $22.2 \pm 8.80$ | |
| | | 0.2 | $83.90 \pm 3.19$ | $\mathbf{84.27 \pm 0.07}$ | $24.14 \pm 10.54$ | |
| | | 0.5 | $\mathbf{88.01 \pm 0.15}$ | $86.42 \pm 0.04$ | $59.22 \pm 0.59$ | |
| | | 1.0 | $\mathbf{88.98 \pm 0.35}$ | $86.77 \pm 0.01$ | $82.12 \pm 3.56$ | |
| | 20 | 0.02 | $\mathbf{82.17 \pm 0.04}$ | $23.34 \pm 0.40$ | $27.97 \pm 2.03$ | 23.83 |
| | | 0.1 | $\mathbf{87.72 \pm 0.14}$ | $63.02 \pm 2.19$ | $26.76 \pm 0.18$ | |
| | | 0.2 | $\mathbf{88.66 \pm 0.06}$ | $82.93 \pm 0.25$ | $36.71 \pm 4.00$ | |
| | | 0.5 | $\mathbf{89.66 \pm 0.09}$ | $84.76 \pm 0.14$ | $50.48 \pm 3.67$ | |
| | | 1.0 | $\mathbf{89.37 \pm 0.17}$ | $80.00 \pm 0.10$ | $84.46 \pm 3.17$ | |

However, our findings indicate that WCE-S2BLK and KL-S2BLK algorithms demonstrate inferior performance compared to WCE-S2BL and KL-S2BL algorithms, which incorporate the logged missing-reward dataset in addition to minimizing the estimation of KL divergence (or reverse-KL divergence). This suggests that the inclusion of the logged missing-reward dataset is beneficial for optimizing KL divergence (or reverse-KL divergence), leading to a more accurate estimation and reduced variance of the IPS estimator.

## H.3    REGRET UPPER BOUND

Using our current theoretical results, we can derive an upper bound on regret, i.e., $|R(\pi_\theta^\star) - R(\pi_\theta^r)|$, where the solution to our regularized risk minimization, denoted by $\pi_\theta^r$.

**Theorem 3.** *Suppose that the reward function takes values in $[-1, 0]$. Then for any $\delta \in (0, 1)$, the following bound on the regret of $\pi_\theta^r(A|X)$ with the truncated IPS estimator (with parameter $\nu \in (0, 0.5]$) holds with probability at least $(1 - 2\delta)$ under distribution $P_X \otimes \pi_0(A|X)$,*

$$|R(\pi_\theta^\star) - R(\pi_\theta^r)| \leq \hat{R}_\nu(\pi_\theta^\star, S) - \hat{R}_\nu(\pi_\theta^r, S) + \frac{4\log(1/\delta)}{3\nu n} + \sqrt{\frac{2\log(1/\delta)M}{n}},$$

*where $M = \min\left(\mathrm{KL}(\pi_\theta^\star \| \pi_0), \mathrm{KL}(\pi_0 \| \pi_\theta^\star)\right) + \min\left(\mathrm{KL}(\pi_\theta^r \| \pi_0), \mathrm{KL}(\pi_0 \| \pi_\theta^r)\right).$*

*Proof.* In Theorem 1, our upper bound holds on true risks of any parameterized policy $\pi_\theta(A|X)$. Therefore, it also holds for optimal $\pi_\theta^\star$ and Therefore, using the following decomposition, we have

$$R(\pi_\theta^\star) - R(\pi_\theta^r) = I_1 + I_2 + I_3,$$

where

$$I_1 := R(\pi_\theta^\star) - \hat{R}_\nu(\pi_\theta^\star, S),$$
$$I_2 := \hat{R}_\nu(\pi_\theta^\star, S) - \hat{R}_\nu(\pi_\theta^r, S),$$
$$I_3 := \hat{R}_\nu(\pi_\theta^r, S) - R(\pi_\theta^r).$$

Note that we have,

$$|R(\pi_\theta^\star) - R(\pi_\theta^r)| \leq |I_1| + I_2 + |I_3|,$$

where we can apply Theorem 1 on $|I_1|$ and $|I_3|$, to provide an upper bound. Therefore, the following upper bound holds on regret of our regularized algorithm with probability at least $1 - 2\delta$,

$$|R(\pi_\theta^\star) - R(\pi_\theta^r)| \leq \hat{R}_\nu(\pi_\theta^\star, S) - \hat{R}_\nu(\pi_\theta^r, S) + \frac{4\log(1/\delta)}{3\nu n} + \sqrt{\frac{2\log(1/\delta)M}{n}},$$

where $M = \min\left(\mathrm{KL}(\pi_\theta^\star\|\pi_0), \mathrm{KL}(\pi_0\|\pi_\theta^\star)\right) + \min\left(\mathrm{KL}(\pi_\theta^r\|\pi_0), \mathrm{KL}(\pi_0\|\pi_\theta^r)\right)$. Therefore, our results can be applied to provide an upper bound on the regret of our algorithm. $\square$

We can observe from Theorem 3, where the upper bound on the regret depends on KL divergence or reverse-KL divergence between the pair $(\pi_\theta^\star, \pi_0)$. Therefore, for a bounded upper bound on regret of our algorithms, we need to have absolutely continuous assumption between these two distributions.

