# OpenReview forum: "Semi-supervised batch learning from logged data"
_ICLR.cc/2024/Conference — ICLR 2024 Conference Withdrawn Submission_

### Official Review · Reviewer_CHNF · 2023-10-29

**Soundness:** 3 good
**Presentation:** 3 good
**Contribution:** 2 fair
**Rating:** 8
**Confidence:** 4

**Summary:**

This paper studies batched policy learning in settings where both labeled and unlabeled data are available, both with known propensities. Based on an observation on the variance of (truncated) inverse-propensity-weighting (IPW) estimators, this paper proposes to penalize the IPW estimator with estimated KL divergence between the policy and base policy which is estimated using missing-label datasets. This idea is instantiated with a concrete algorithm with gradient descent, and extensive experiments are conducted to demonstrate the superior performance of the proposed method compared with other competitors.

**Strengths:**

1. This paper contributes to a significant research question, i.e., policy learning from logged data. It contributes to the existing literature by introducing a KL-regularization framework, and proposing to estimate KL with unlabeled data (while still being powerful when KL is in-sample estimated).

2. Originality. The idea of regularization is not new (in fact there is some missing literature I'll discuss in the Weaknesses section), but the method presented here contains new techniques and contributes new insights.

2. The quality and clarity of the paper is good overall. This paper is well organized and clearly presented. It contains fruitful results covering theory, algorithm, and experiments.

**Weaknesses:**

1. Related literature.

In general this paper does a good job of relating to existing literature, but the idea of penalizing with KL divergence is closely related to the literature on "Pessimism" for policy learning, e.g., [1] in offline RL, where value estimation is penalized with uncertainty estimation (related to how close a target policy is to the behavior policy), or [2] which is closer to the batched policy learning problem. The point is that by penalization and optimizing a lower confidence bound, one can achieve better performance (which the framework of Joachims et al, 2015 cannot due to truncation). I am curious whether the results in this paper can be useful in deriving a different LCB for that case. Maybe it is impossible to do so given the space limit, but some discussion can be interesting.

[1] Jin, Y., Yang, Z. and Wang, Z., 2021, July. Is pessimism provably efficient for offline rl?. In International Conference on Machine Learning (pp. 5084-5096). PMLR.
[2] Jin, Y., Ren, Z., Yang, Z. and Wang, Z., 2022. Policy learning" without''overlap: Pessimism and generalized empirical Bernstein's inequality. arXiv preprint arXiv:2212.09900.

2. Evaluation metric in experiments.

The paper could be improved by being more explicit on the evaluation metric -- is that the value of learned policy, as I understand the target as learning an optimal policy? It is a bit unclear to me what "accuracy" means in this context.

3. benchmark in experiments.

Is BanditNet the only competitor (in terms of complex learners) in this literature? At least there should be a discussion on this point.

**Questions:**

1. Is unlabeled data so important for estimating KL?

It seems the only use of unlabeled data is to estimate KL (imputation is ruled out with solid discussion which I appreciate), but even if the KL is estimated with labeled data, the performance is fine. I guess the reason is that the estimation error of KL is secondary in Equation (8) because of a $1/n$ factor. Labeled data should be able to learn KL with $O(1/\sqrt{n})$ error which doesn't really hurt?

2. How does this method relate to pessimism-based methods?

This is the same as Weakness point 1.

3. Two questions on experiments in Weakness part.

---

> ### Author Response · Authors · 2023-11-17
> **Response [1/2]**
>
> We appreciate Reviewer CHNF for taking the time to read our work and to share helpful feedback. We are glad that Reviewer  CHNF found our work solid, from both the theoretical perspective and the algorithmic perspective. We respond to the reviewer's comments below.
>
> ## Application of our theoretical approach in Pessimism policy learning
>
> > Related literature. In general this paper does a good job of relating to existing literature, but the idea of penalizing with KL divergence is closely related to the literature on "Pessimism" for policy learning, e.g., [1] in offline RL, where value estimation is penalized with uncertainty estimation (related to how close a target policy is to the behavior policy), or [2] which is closer to the batched policy learning problem. The point is that by penalization and optimizing a lower confidence bound, one can achieve better performance (which the framework of Joachims et al, 2015 cannot due to truncation). I am curious whether the results in this paper can be useful in deriving a different LCB for that case. Maybe it is impossible to do so given the space limit, but some discussion can be interesting.
>
>  In pessimism approach [2], a lower confidence bound (LCB) is assumed as regularization. In particular, the authors assume a policy-dependent and data-dependent regularization, which is also a LCB on $|R(\pi_{\theta})-\hat{R}(\pi_\theta,S)|$. However, in our method, we motivate the regularization by reduction in the variance of (truncated) IPS estimator.
>
> Regarding deriving a different LCB using our method, it is possible to utilize Bernstein’s inequality to provide an upper bound $|R(\pi_{\theta})-\hat{R}(\pi_\theta,S)|$. In particular,  via Proposition 1 we can bound the variance of IPS (or IPW in [2]) with the KL-divergence or reverse KL-divergence under sub-Gaussian assumption which is more relaxed with respect to truncated version. Then, we need to combine Proposition 1 with Bernstein’s inequality in order to derive an upper bound on $|R(\pi_{\theta})-\hat{R}(\pi_\theta,S)|$. Note that, for doubly robust estimators (or augmented-inverse-propensity-weighted (AIPW)-type estimators), we need also to utilize the approach in Proposition 2.
>
> ## On evaluation metric,
>
> > - Evaluation metric in experiments.
> The paper could be improved by being more explicit on the evaluation metric -- is that the value of the learned policy, as I understand the target as learning an optimal policy? It is a bit unclear to me what "accuracy" means in this context.
>
>  The experiments were conducted on the single-class classification task and the accuracy means the proportion of correctly classified samples in the dataset. For a given data sample, correct classification induces reward 1 and misclassification induces reward 0. In single-class classification, accuracy is equivalent to average reward under the argmax strategy, which is also defined in [7].
>
> While accuracy can be helpful in single-class classification, for real-world problems where we don't have the reward of other actions, it is not measurable. Hence, for the real-world experiment that we add, we use empirical IPS based on test dataset as an estimation of the average reward and a measure of the performance of different methods.
>
> ## On BanditNet,
>
> > - benchmark in experiments.
> Is BanditNet the only competitor (in terms of complex learners) in this literature? At least there should be a discussion on this point.
>
>  To the best of our knowledge, BanditNet is the only implementation of Batch learning from logged bandit feedback, [3], via deep learning, based on the self-normalized estimator. Therefore, we have chosen this method as a baseline for our experiments in deep models. In addition, it is assumed that we have access to propensity scores. Note that, other corrections to the IPS estimator, e.g., [4,5,6], can also be implemented by deep learning provided that their objective function is convex. We clarify this discussion in the main body of the paper.
>
> ---

---

> ### Author Response · Authors · 2023-11-17
> **Response [2/2]**
>
> ## On effect of logged missing-reward dataset,
>
> > Is unlabeled data so important for estimating KL?
> It seems the only use of unlabeled data is to estimate KL (imputation is ruled out with solid discussion which I appreciate), but even if the KL is estimated with labeled data, the performance is fine. I guess the reason is that the estimation error of KL is secondary in Equation (8) because of a $1/n$ factor. Labeled data should be able to learn KL with $O(1/\sqrt{n})$ error which doesn't really hurt?
>
> Great point!
>
> - From an experimental perspective, in Section F.9 (Appendix), we conducted some experiments to investigate the effect of logged missing reward datasets. For this purpose, we fixed the number of logged known-reward samples and then increased the number of logged missing-reward data samples. We can observe that the logged missing-reward dataset can help us by improving the accuracy. Therefore, the logged missing-reward (unlabeled) are important in estimating KL divergence.
>
> - From a theoretical perspective, as we have
>
> $$\mathcal{O}\Big(\big( \min( \mathrm{KL} (\pi_\theta \| \pi_0 ) , \mathrm{KL} ( \pi_0 \| \pi_\theta ) ) \big)^{1/4} / \sqrt{n} \Big)$$
>
> in Equation (8), then it would result in convergence rate of $\mathcal{O}(1/\sqrt{n})$, provided that the $\min(\mathrm{KL}(\pi_\theta\|\pi_0),\mathrm{KL}(\pi_0\|\pi_\theta))$ is bounded. Therefore, the estimation of KL divergence is important. Note that, in Proposition 5, we discussed an upper bound on our estimators of KL-divergence and reverse-KL divergence, where it shows that the we can learn KL divergence or reverse-KL divergence with the rate $\sum_{i=1}^k \sqrt{\frac{1}{m_{a_i}}}$ where $\sum_{i=1}^k m_{a_i}=m+n$ and $m$ is the number of logged-missing reward samples. Based on Proposition 5, we can observe that the logged missing reward samples are helpful in the estimation of KL-divergence or reverse-KL divergence.
>
> ## Compare with pessimism-based methods,
>
> > How does this method relate to pessimism-based methods?
>
>  These are the differences,
>
> - **Motivation of regularization:** The main difference is the motivation for regularization (or penalty). In our method, the motivation for regularization is based on reducing the variance of the truncated IPS estimator, similar to [3]. However, in pessimistic methods, the regularization is motivated by LCB bounds on the estimation uncertainty of the augmented-inverse-propensity-weighted (AIPW)-type estimators which also include IPS estimators.
>
> - **Computationally tractable and convexity:** Our regularization, KL-divergence or reverse-KL divergence, is convex with respect to parameterized learned policy, $\pi_\theta$, and is computationally tractable. However, in [2], as discussed in their Discussion section (Section 7), one of their extension to their approach is developing computational tractable version of their algorithms. As future work, it is interesting to apply our methods in pessimistic methods, e.g., [1,2], and derive computational tractable algorithms.
>
> - **Behavior policy:** In the pessimistic method, it is assumed that the logging policy (or behavior policy) is known, e.g., [2, Assumption 2.1]. However, in [3] or our work, it is assumed that the logging policy is partially known (or unknown).
>
> - **Unbounded loss and sub-Gaussianity assumption:** In the pessimistic method, the unbounded loss (IPS estimator) is studied, [2]. In our case, we studied the truncated IPS estimator. However, in our proposition 1, our result holds for the IPS estimator under the sub-Gaussianity assumption (which is relaxed with respect to the truncated assumption).
>
> Finally, we added related works about the pessimistic method in other related work section (Appendix A).
>
> ---
>
> **References:**
>
> - [1] Jin, Y., Yang, Z. and Wang, Z., July. Is pessimism provably efficient for offline rl?. ICML 2021.
>
> - [2] Jin, Y., Ren, Z., Yang, Z. and Wang, Z., 2022. Policy learning" without''overlap: Pessimism and generalized empirical Bernstein's inequality. arXiv preprint arXiv:2212.09900.
>
> - [3]: Adith Swaminathan and Thorsten Joachims. Batch learning from logged bandit feedback through counterfactual risk minimization. JMLR, 2015a.
>
> - [4]: Miroslav Dudík, John Langford, and Lihong Li. Doubly robust policy evaluation and learning. ICML, 2011.
>
> - [5]:Alberto Maria Metelli, Alessio Russo, and Marcello Restelli. Subgaussian and differentiable importance sampling for off-policy evaluation and learning. Neurips, 2021.
>
> - [6]: Imad Aouali, Victor-Emmanuel Brunel, David Rohde, and Anna Korba. Exponential smoothing for off-policy learning. In ICML, 2023.
>
> - [7]: London, B. and Sandler, T., 2019, May. Bayesian counterfactual risk minimization. In ICML.

---

> > ### Comment · Reviewer_CHNF · 2023-11-20
> >
> > Thank you for your clarification on the related work, evaluation metric and benchmarking methods, which is helpful. For the estimation of KL: I still think that the estimation error is a smaller order term compared with the constant KL divergence itself, so a consistent estimator suffices. But still, it's helpful to include such discussion in the paper. I raise my score to 7.

---

> > > ### Author Response · Authors · 2023-11-20
> > > **Response**
> > >
> > > We thank Reviewer CHNF for the helpful feedback.
> > >
> > > >For the estimation of KL: I still think that the estimation error is a smaller order term compared with the constant KL divergence itself, so a consistent estimator suffices.
> > >
> > > We would like to provide more insights from an experimental perspective. For example, suppose that we have mini-batches of size 128, and the reward is known in 2\% of the training dataset, only 3 known-reward samples would be used to estimate the KL divergence, which is not sufficient. Therefore, missing-reward samples would be helpful.
> > >
> > > >But still, it's helpful to include such a discussion in the paper.
> > >
> > > If you agree, we will add this discussion to the paper following the collection of change requests from other reviewers.
> > >
> > > >*I raise my score to 7*.
> > >
> > > We appreciate your support of our work.
> > > The feedback you provided inspired us to implement our techniques in pessimistic reinforcement learning—an intriguing domain within RL.

---

> > > ### Author Response · Authors · 2023-11-22
> > > **Using known-reward dataset for estimation of KL and reverse KL**
> > >
> > > To enhance our understanding of KL-divergence and reverse KL-divergence estimation using logged known-reward dataset (labeled dataset), we present some insightful experimental findings.
> > >
> > > We conducted some experiments for WCE-S2BL and KL-S2BL using the logged known-reward dataset for regularization. These algorithms are referred to as *WCE-S2BLK* and *KL-S2BLK*, respectively. The results for these algorithms, i.e., *WCE-S2BLK* and *KL-S2BLK*, are shown in Tables 1 and 2 in the main body and Tables 6-9 in Appendix F.
> > >
> > > We can observe that these algorithms' results (*WCE-S2BLK* and *KL-S2BLK*) are worse than those for *WCE-S2BL* and *KL-S2BL* where both logged known-reward and missing reward datasets are utilised in the estimation of KL divergence and reverse-KL divergence. For example, in Table 2 for CIFAR-10 in deep model, $\tau=1$ and $\rho=0.02$,  we have,
> > >
> > >
> > > | $\tau$ | $\rho$ |  *WCE-S2BL* | *KL-S2BL* | *WCE-S2BLK* | *KL-S2BLK* |
> > > | -----------| -----------| ----------- | ----------- | ----------- | ----------- |
> > > | 1 | 0.02 | $85.01 \pm 0.37$      |    $84.6 \pm 0.65$    | $17.12 \pm 0.97$ | $21.63 \pm 1.44$|
> > > | 1 | 0.2 | $85.06\pm 0.32$ | $85.53\pm 0.56$ | $58.04\pm 5.47$ | $54.12\pm 0.51$|
> > >
> > > where the accuracies of *WCE-S2BLK* and *KL-S2BLK* are lower than *WCE-S2BL* and *KL-S2BL*, respectively. Therefore, the logged missing-reward dataset can be helpful in the estimation of KL divergence or reverse-KL divergence.
> > >
> > > ---
> > >
> > > We are happy to discuss more and answer any additional questions the reviewer CHNF might have.

---

### Official Review · Reviewer_G8No · 2023-11-02

**Soundness:** 3 good
**Presentation:** 3 good
**Contribution:** 2 fair
**Rating:** 6
**Confidence:** 4

**Summary:**

This paper addresses the problem of off-policy learning using logged bandit data, where some data have unobserved action rewards, called missing-rewards data. To handle this problem, the authors first derive an upper bound on the true risk under the inverse propensity score estimator in terms of divergences between the logging policy and a parameterized policy. These KL divergences are independent of the reward function. The authors leverage this to propose a learning algorithm with reward-free regularization which leverages the
availability of the logged known-reward data and the logged missing-reward data.  For the practical algorithm, they also propose a consistent and asymptotically unbiased estimator of KL divergence and reverse KL divergence between the logging policy and a parameterized policy. Finally, the effectiveness of their proposed algorithm is demonstrated through simulation experiments, demonstrating their advantages.

**Strengths:**

- This paper is well-written and easy to follow. The theoretical definitions and most of the related works are clearly explained.
- The approach and the derived upper bound for the true risk are original.
- The experiments are enough to demonstrate their approach.

**Weaknesses:**

- Although the derived upper bound is new, the approach for the learning algorithm to leverage this bound is quite straightforward. In addition, the practical algorithm for their approach is lack of a theoretical guarantee (i.e.,  an upper bound for the regret).

- The experiments only include two datasets. It would be better to add a real-life example in the bandit setting.

**Questions:**

- The authors mention that the direct method where we estimate the reward function fails to generalize well. What happens if we use a neural network to estimate the reward function?

- Compared to the CRM framework (Swaminathan & Joachims, 2015a) which uses empirical variance minimization, they minimize the divergence KL. Could the authors explain the advantages of using this divergence KL compared to the empirical variance?

- In Table 1, why does the proposed algorithm significantly outperform the logging policy for the FMNIST dataset?

---

> ### Author Response · Authors · 2023-11-17
> **Respones [1/2]**
>
> We appreciate Reviewer G8No for taking the time to read our work and to share helpful feedback. We are glad that Reviewer G8No found the approach and the derived upper bound original. Reviewer G8No's comments and pointers have been used to improve our work. We respond to the reviewer's comments below.
>
> ## Regret bound discussion
>
> > Although the derived upper bound is new, the approach for the learning algorithm to leverage this bound is quite straightforward. In addition, the practical algorithm for their approach is lack of a theoretical guarantee (i.e., an upper bound for the regret).
>
>  Using our current theoretical results, we can derive an upper bound on regret, i.e., $|R(\pi^\star_\theta)- R(\pi^r_{\theta})|$. In Theorem 1, our upper bound holds on true risks of any parameterized policy $\pi_\theta(A|X)$. Therefore, it also holds for optimal $\pi^\star_{\theta}$ and the solution to our regularized risk minimization, denoted by $\pi^r_{\theta}$. Therefore, using the following decomposition, we have,
>
> $R(\pi^\star_\theta)- R(\pi^r_{\theta}) = I_1 + I_2 +I_3,$
>
> where
>
> $$
> I_1:= R ( \pi^\star_\theta) - \hat{R_\nu} (\pi^\star_\theta , S),
> $$
>
> $ I_2:= \hat{R_\nu} (\pi^\star_\theta,S)- \hat{R_\nu} (\pi^r_{\theta},S), $
>
> $ I_3:= \hat{R_\nu} (\pi^r_{\theta},S) -R (\pi^r_{\theta}). $
>
> Note that we have,
>
> $|R(\pi^\star_\theta)- R(\pi^r_{\theta})| \leq |I_1| + I_2 + |I_3|,$
>
> where we can apply Theorem 1 on $|I_1|$ and $|I_3|$, to provide an upper bound. Therefore, the following upper bound holds on regret of our regularized algorithm with probability at least $1-2\delta$,
>
>  $ | R(\pi^\star_\theta)- R(\pi^r_{ \theta }) |  \leq \hat{R_\nu} ( \pi^\star_\theta , S) - \hat{R_\nu} (\pi_{\theta}^r , S)  + \frac{ 4 \log(1/\delta)}{ 3\nu n } + \sqrt{ \frac{2\log( 1/\delta ) M }{ n }}, $
>
> where $M=\min\big(\mathrm{KL}(\pi^\star_\theta\|\pi_0),\mathrm{KL}(\pi_0\|\pi^\star_\theta)\big)+\min\big(\mathrm{KL}(\pi^r_{\theta}\|\pi_0),\mathrm{KL}(\pi_0\|\pi^r_{\theta})\big)$. Therefore, our results can be applied to provide an upper bound on the regret of our algorithm.
>
> ## Real-life example,
>
> > The experiments only include two datasets. It would be better to add a real-life example in the bandit setting.
>
>  We added experiments on **KuaiRec** which is a real-world dataset of about 16M human interactions with recommendation logs of a video-sharing app. The details of the implementation and results are available in Appendix F.7 and Table 11 respectively.
>
> ## On direct method,
>
> > The authors mention that the direct method where we estimate the reward function fails to generalize well. What happens if we use a neural network to estimate the reward function?
>
>  Good point! In the direct method (reward function estimation), we need to first estimate the reward function and then at the second stage, we need to employ a supervised learning algorithm to design the optimal policy. Our approach directly tackles the problem using KL-regularized (or reverse-KL-regularized) truncated IPS estimator instead, using computation and space complexity that mimics supervised approaches to the problem.
>
> Note that, in our method, we can reduce the variance of the IPS estimator via utilizing the KL-divergence (or reverse-KL divergence) regularization. In fact, we can provide an upper bound in estimation uncertainty in our method. However, in direct method, we can not quantify the estimation uncertainty.
>
> Furthermore, when estimating the reward function via a neural network, a substantial quantity of logged known-reward samples is required. However, in practical scenarios, the availability of such logged known-reward datasets is often limited.
>
> ## KL-divergence ( reverse-KL divergence) vs empirical variance,
>
> > Compared to the CRM framework [1] which uses empirical variance minimization, they minimize the divergence KL. Could the authors explain the advantages of using this divergence KL compared to the empirical variance?
>
>  It is worth mentioning that the regularization based on empirical variance proposed by [1] depends on rewards and we can not minimize the empirical variance via logged missing-reward dataset. Therefore, we proposed KL-divergence (or reverse KL-divergence) which is reward-independent and can be minimized via both logged known-reward and missing-reward datasets. In addition, our estimators of KL divergence and reverse-KL divergence are convex and we can be implemented via deep models. Note that the empirical variance is not convex.
>
> ---

---

> > ### Author Response · Authors · 2023-11-17
> > **Response [2/2]**
> >
> > ## FMNIST discussion,
> >
> > > In Table 1, why does the proposed algorithm significantly outperform the logging policy for the FMNIST dataset?
> >
> >  Great point. During the supervised-to-bandit dataset process, we utilized the raw images of FMNIST for features. FMNIST is a relatively simple image dataset which can be modeled accurately by a linear model in the bandit-feedback setting. In the experiments for $\tau=10$ we use a logging policy which is a smoothed version of a linear policy that has 91\% accuracy, hence it still can achieve higher accuracy with softmax policy. Despite of good performance of logging policy in $\tau=1$, our algorithms' performance (accuracy) is better than the baseline, i.e., B-CRM. Note that, our model is also able to keep high accuracy even with 2\% of the known-reward data samples, but it also uses the remaining 98\% of the data with missing-reward, utilizing the information available in the logging policy.
> >
> > ---
> >
> > **References :**
> >
> > - [1]: Adith Swaminathan and Thorsten Joachims. Batch learning from logged bandit feedback through
> > counterfactual risk minimization. The Journal of Machine Learning Research, 16(1):1731–1755,
> > 2015a.

---

> > > ### Author Response · Authors · 2023-11-22
> > > **Thanks for your time and consideration**
> > >
> > > Dear Reviewer G8No,
> > >
> > > We would like to thank you again for your comments! We hope our response has addressed all your concerns. Since we are approaching the end of the discussion period, please let us know if you have any other questions, and we are happy to discuss more.
> > >
> > > Bests,
> > > Authors

---

### Official Review · Reviewer_KCYk · 2023-11-07

**Soundness:** 3 good
**Presentation:** 2 fair
**Contribution:** 3 good
**Rating:** 5
**Confidence:** 4

**Summary:**

This work addresses the problem of batch off-line learning for contextual bandits when some of the rewards are missing from the logged dataset. The proposed approach comprises of two components: augmenting the logged dataset by predicting the missing rewards in the logged dataset using an outcome model trained on the observed samples, and estimating the importance weights on both the missing and non-missing reward observations (contexts and actions are not assumed missing). Experimental results show strong performance of the proposed estimator.

**Strengths:**

Batch off-policy learning is a common and important problem in practice. The authors identify a particularly interesting setting where rewards are missing from the responses. The use of the importance sampling weights is well motivated by noting the relationship between the importance weights and the risk of the respective policies. The use of pseudo labels is also an interesting approach.

**Weaknesses:**

My main issue with this work is that the presentation is such that it is difficult to parse the contribution of the work. The authors place great emphasis on their result on relating the risks to the KL-divergence (an other Bregman divergences), but then also introduce methodology without a lot of details. It's also not entirely clear to me which assumptions are being employed here in order to make the method applicable. The authors note that access to the true logging policy, however it would also seem that at a minimum there is also necessary conditions on overlap. It also isn't entirely clear to me what assumptions are being made on the missingness process. Are we assuming that the reward is missing completely at random here? If not, there would seem to be an issue with a biased estimate of the missing rewards.

**Questions:**

I inlined a few questions above, but it would be great if the authors could more completely describe the assumptions here and where they need to be employed in the theoretical results.

* Given that you are augmenting missing observations with an outcome model, it would seem that a relevant comparison would be a Q learning approach for off-policy optimization.

* It would seem that the authors should also discuss some of the work on causal inference with missing outcomes, e.g. https://arxiv.org/abs/2201.00468 and https://arxiv.org/abs/2305.12789

---

> ### Author Response · Authors · 2023-11-17
> **Response [1/2]**
>
> We appreciate the reviewer KCYk taking the time to read our work and to share helpful feedback. Your comments and pointers have been used to improve our work. With respect to the weaknesses which were pointed out:
>
> > My main issue with this work is that the presentation is such that it is difficult to parse the contribution of the work. The authors place great emphasis on their result on relating the risks to the KL-divergence (an other Bregman divergences), but then also introduce methodology without a lot of details.
>
>  The main concern in IPS estimators is the high variance of IPS estimator. We proved that the minimization of KL divergence (or reverse-KL divergence) would result in variance reduction of (truncated) IPS estimator, Proposition 1. Then, we provide an upper bound on the true risk of IPS estimator which is dependent on KL-divergence. Finally, we discuss that these regularization terms are reward-free and can be minimized via both logged known-reward and missing-reward datasets. We also introduce novel estimators for the KL divergence (or reverse-KL divergence).
>
> ## Assumptions
>
> > It's also not entirely clear to me which assumptions are being employed here in order to make the method applicable. The authors note that access to the true logging policy, however it would also seem that at a minimum there is also necessary conditions on the overlap.
>
>  Thanks for your helpful comment. In our problem formulation, we mentioned that the logging (behavioural) policy is partially known and we do not have access to logging policy.
>
> Regarding the overlap assumption,
>
> - From theoretical perspective, we consider the truncated IPS estimator where we do not need overlap assumption, as we have bounded variance. In addition, to have non-vacuous upper bound we need to assume that the KL-divergence or reverse-KL divergence are bounded (due to absolutely continuous assumption). Therefore, we have a relaxed version of overlap assumption in this case.
>
> - From experiment perspective, since we're considering the logging policy and a parameterized learned policy within the Softmax policy class in our experiments, the overlap assumption  automatically holds.
>
> ## Missing-reward dataset distribution
>
> > It also isn't entirely clear to me what assumptions are being made on the missingness process. Are we assuming that the reward is missing completely at random here? If not, there would seem to be an issue with a biased estimate of the missing rewards.
>
>  In this work, we assume that the reward is missed completely randomly. Actually, we assume that the distribution of missing-reward dataset is the same as known-reward dataset. It is mentioned before section 4: "In our S2BL setting, we also have access to the logged missing-reward dataset, which we shall denote as $S_{u}=(x_j,a_j,p_j)_{j=1}^m$ and assume it is generated independent and identically distributed under the same logging policy for the logged known-reward dataset, i.e., $p_j=\pi_0(a_j|x_j)$." The extension of our results for selection bias between missing-reward and known-reward datasets, as discussed in [1], is achievable and we consider as our future plan.
>
> > I inlined a few questions above, but it would be great if the authors could more completely describe the assumptions here and where they need to be employed in the theoretical results.
>
>  Thanks for your suggestion. We clarified our assumptions in the main body of the paper.
>
> ---

---

> > ### Author Response · Authors · 2023-11-17
> > **Response [2/2]**
> >
> > ## Q-learning
> >
> > > Given that you are augmenting missing observations with an outcome model, it would seem that a relevant comparison would be a Q-learning approach for off-policy optimization.
> >
> >  Thanks for your suggestion. It is worthwhile to mention that in our approaches, we do not use Pseudo-label (or pseudo-rewards). If we want to utilize the pseudo-label, then we need to estimate the reward function which is the direct method. Therefore, we do not augment missing-reward with pseudo-labels generated from an outcome model. Note that our datasets also contain propensity scores, which are unavailable in off-policy reinforcement learning.
> >
> > ## Causal inference with missing outcomes
> >
> > > It would seem that the authors should also discuss some of the works on causal inference with missing outcomes, e.g. [1] and [2].
> >
> >  Thanks for introducing interesting works on causal inference with missing outcomes. We discussed these works at other related works in "Individualized Treatment Effects" paragraph. Note that, our work differs from these works, due to, limited the action space which contains *two actions* in these works. In [1], a doubly-robust mean-estimation under semi-supervised learning is studied which is similar to direct method.
> >
> > ---
> >
> > **References:**
> >
> > - [1]: Zhang, Y., Chakrabortty, A. and Bradic, J., 2023. Semi-Supervised Causal Inference: Generalizable and Double Robust Inference for Average Treatment Effects under Selection Bias with Decaying Overlap. arXiv preprint arXiv:2305.12789.
> >
> > - [2]: Chakrabortty, A., Dai, G. and Tchetgen, E.T., 2022. A general framework for treatment effect estimation in semi-supervised and high dimensional settings. arXiv preprint arXiv:2201.00468.

---

> ### Author Response · Authors · 2023-11-22
> **Q-learning Experiments**
>
> ## Q-learning experiment
>
> To assess the relative performance of our methods, WCE-S2BL and KL-S2BL, we tried to conduct experiments comparing them to Q-learning.
>
> In Q-learning approach, we first **estimate the reward function** using a logged known-reward dataset, employing logistic regression with a sigmoid activation function and a linear layer. Note that, in this scenario the rewards are binary. Second, we generate **pseudo-rewards** by applying the reward function estimator to the logged missing-reward dataset. Finally, we train the **truncated IPS estimator** with both the logged known-reward dataset and the pseudo-reward dataset.
>
> In the following table, we present the results (accuracy) of our algorithms (WCE-S2BL and KL-S2BL) and Q-learning under the EMNIST dataset with varying ratios of missing-reward data to known-reward data.
>
> | $\tau$ | $u/l$ ratio | *WCE-S2BL* | *KL-S2BL* | Q-learning |
> | -----------| -----------| ----------- | ----------- | ----------- |
> | 1 | 9 | $87.52   \pm 0.00$      |    $69.79 \pm 0.56$     | $22.34 \pm 0.48$ |
> | 1 | 49 | $87.00  \pm 0.01$      | $77.18\pm 0.37$        | $26.16\pm 1.30$ |
> | 20 | 9 | $87.72  \pm 0.14$      |    $63.02 \pm 2.19$    | $26.76 \pm 0.18$ |
> | 20 | 49 | $82.17 \pm 0.04$     | $33.34 \pm 0.40$       | $27.97\pm 5.47$ |
>
> As we can observe, our method (WCE-S2BL) has a better accuracy in comparison with Q-learning.
>
> ---
>
> Thanks again for your comments! We hope our response has addressed all your concerns. Since we are approaching the end of the discussion period, please let us know if you have any other questions, and we are happy to discuss more.

---

### Official Review · Reviewer_p1oi · 2023-11-08

**Soundness:** 4 excellent
**Presentation:** 2 fair
**Contribution:** 2 fair
**Rating:** 5
**Confidence:** 3

**Summary:**

They provide a semi-supervised method for logged data with missing rewards. Their method is built on top of the IPS policy gradient methods, with regularization terms. The regularization term is the KL divergence or the reverse KL divergence between the learned policy and the logging policy, which basically constrains the learned policy not too far away from the logging policy.

They propose an upper bound on true risk under the IPS estimator in terms of different divergences.

**Strengths:**

1, The idea is straightforward and well-motivated.
2, Theorem 1 bound the IPS reward on all samples by the known reward samples, which build the foundation of their proposed algorithm.
3, Experiments result of the neural network policy is good and the improvement compared to the baseline is huge.

**Weaknesses:**

1, The paper writing is not clear enough.
2, The proposed method lacks novelty. This is a policy constraint method.
3, Doesn't explain what's the advantages of their method when dealing with the missing reward samples.
4, The writing of introduction only introduces the background and doesn't include the motivation of their method. Also, the logic of the intro and the related work is not good enough. For example, in section 5, the Pseudo-labeling algorithm part is confusing, which makes me believe that you are studying a Pseudo-labeling algorithm. I think they should move this part to the intro or related work.
5, The experiment is not convincing since they only have 2 datasets. Also, the linear case has bad results. The cifar-10 dataset is bad when using a linear classifier in supervised learning, so I wonder why they use a linear policy.
6, They assume the logging policy is known, which is not always practical. And we all know that the policy constraint methods suffer from the estimation of the logging policy.

**Questions:**

1, When to use WCE and when to use KL divergence?
2, How to choose v in to truncate the importance weight?

---

> ### Author Response · Authors · 2023-11-17
> **Response [1/2]**
>
> We appreciate Reviewer p1oi taking the time to read our work and to share helpful feedback. We respond to the reviewer's comments below.
>
> ## on Policy Constraint,
> >  The proposed method lacks novelty. This is a policy constraint method.
>
>   In constrained policy optimization [3], we are searching for the optimal policy within a set $\Pi_{\theta}\subset \Pi$ of parametrized policies with parameters $\theta$. For this purpose, the optimization is done over a local neighborhood of the most recent iterate policy measured via a distance, i.e.,
> $$\min_{\pi_{\theta}^k}\hat{R}(\pi_\theta,S), \quad \text{s.t.} \quad D(\pi_{\theta}^k,\pi_{\theta}^{k-1})\leq \delta, $$
> where $D(\cdot,\cdot)$ is a distance measure, e.g., total variation distance. Then, by applying the Pinsker inequality, the constraint would be in terms of the square root of KL divergence between successive policies during parametric policy iteration to avoid large steps. However, we differ from constraint policy optimization, due to,
>
> - We motivate the KL-regularization (or reverse-KL regularization) from variance reduction of truncated IPS estimator which is different from policy constraint approach as discussed.
>
> - Our divergences are between the parameterized policy at each iteration and the logging policy. However, in constrained policy optimization, the KL divergence or distance measure is computed between two successive policies during policy iterations.
>
> - In addition, we also consider the reverse-KL regularization, WCE-S2BL algorithm, which is different from the common divergence in constraint policy method, which is KL-divergence.
>
> ## Advantage of logged missing-reward dataset,
>
> > Doesn't explain what's the advantages of their method when dealing with the missing reward samples.
>
> The main advantage of our method is the novel estimator of KL divergence or reverse KL divergence. These estimators can be implemented by applying both logged known-reward and missing-reward datasets. It is discussed in Section 5.
>
> > The writing of introduction only introduces the background and doesn't include the motivation of their method. Also, the logic of the intro and the related work is not good enough. For example, in section 5, the Pseudo-labeling algorithm part is confusing, which makes me believe that you are studying a Pseudo-labeling algorithm. I think they should move this part to the intro or related work.
>
>  Thanks for your suggestion, we moved this discussion to other related works in Appendix A.
>
> ## New experiments,
>
> > The experiment is not convincing since they only have 2 datasets. Also, the linear case has bad results. The CIFAR-10 dataset is bad when using a linear classifier in supervised learning, so I wonder why they use a linear policy.
>
>  We added experiments for extra two datasets, **EMNIST** and **CIFAR-100**, as well as **KuaiRec** which is real-world dataset of human interactions. The details of the added datasets are available in Appendix F. The results of EMNIST and CIFAR-100 are added in Tables 7 and 9. Details of the KuaiRec dataset are explained in Appendix F.7 and results on this dataset are available in Table 11.
>
> In the linear experiments for CIFAR-10 and FashionMNIST, the model is trained based on logged datasets from a deep logging policy. Due to the fact that the complexity of the logging policy as a deep model is more than a linear model, the linear CIFAR-10 model accuracy is worse than the logging policy. The reason behind this setting is that a simple linear model doesn't work well on the raw flattened image and pre-trained features inject unknown prior information into the input of the models. To observe the difference between these two settings, in the experiments for the added datasets, we use a linear model for both the logging policy and the trained policy, using pre-trained features as image representation. We also added another experiment for pre-trained features of CIFAR-10 in Appendix F.6, to observe the difference on the same dataset.
>
> ---

---

> ### Author Response · Authors · 2023-11-17
> **Response [2/2]**
>
> ## On logging policy,
>
> > They assume the logging policy is known, which is not always practical. And we all know that the policy constraint methods suffer from the estimation of the logging policy.
>
>  Regarding to assumption on known logging policy,
>
> - Note that, inspired by [1], we assume that the **logging policy is partially unknown** and we just have access to some samples of logging policy ( i.e., propensity scores) and we neither know the parameters and architecture of the logging policy nor can query from the logging policy. Therefore, we do not assume the logging policy is known.
>
> Regarding to estimation of propensity scores, note that,
>
> - In scenarios where we have no access to propensity scores, we can estimate the propensity score using different methods. For example, we can estimate the propensity score using logistic regression [4,5], generalized boosted models [6], neural networks [7], parametric models [10] or
> classification and regression trees [8].
>
> - In addition, there is an evidence [9], [11, Chapter 9, Page 206] and [12] that using estimated propensities scores offers a better bias-variance trade-off than using the true propensities. Therefore, in scenarios with unknown propensity scores, the variance under estimated propensity scores would be even smaller.
>
> ## Response to Questions,
>
> >  When to use WCE and when to use KL divergence?
>
>  From the experiments on different datasets, i.e., FMNIST, CIFAR-10, CIFAR-100, EMNIST and KuaiRec, we can observe that WCE-S2BL algorithm is more robust and has a better performance (accuracy) in comparison with KL-S2BL algorithm.
>
> >  How to choose $\nu$ in to truncate the importance weight?
>
>  Inspired by BanditNet experiments in [2], for the FMNIST, EMNIST, CIFAR-10 and CIFAR-100 datasets, we consider the truncation hyperparameter $\nu=0.001$.
>
> ----
> **References:**
>
> - [1]: Adith Swaminathan and Thorsten Joachims. Batch learning from logged bandit feedback through
> counterfactual risk minimization. The Journal of Machine Learning Research, 16(1):1731–1755,
> 2015a.
>
> - [2]: Thorsten Joachims, Adith Swaminathan, and Maarten de Rijke. Deep learning with logged bandit
> feedback. In International Conference on Learning Representations, 2018.
>
> - [3]: Achiam, J., Held, D., Tamar, A. and Abbeel, P., 2017, July. Constrained policy optimization. In International conference on machine learning (pp. 22-31). PMLR.
>
> - [4]: Ralph B D’Agostino Jr. Propensity score methods for bias reduction in the comparison of a treatment
> to a non-randomized control group. Statistics in Medicine, 17(19):2265–2281, 1998.
>
> - [5]: Sherry Weitzen, Kate L Lapane, Alicia Y Toledano, Anne L Hume, and Vincent Mor. Principles for
> modeling propensity scores in medical research: a systematic literature review. Pharmacoepidemi-
> ology and Drug Safety, 13(12):841–853, 2004.
>
> - [6]: Daniel F McCaffrey, Greg Ridgeway, and Andrew R Morral. Propensity score estimation with
> boosted regression for evaluating causal effects in observational studies. Psychological methods, 9
> (4):403, 2004.
>
> - [7]: Soko Setoguchi, Sebastian Schneeweiss, M Alan Brookhart, Robert J Glynn, and E Francis Cook.
> Evaluating uses of data mining techniques in propensity score estimation: a simulation study.
> Pharmacoepidemiology and Drug Safety, 17(6):546–555, 2008.
>
> - [8]: Brian K Lee, Justin Lessler, and Elizabeth A Stuart. Improving propensity score weighting using
> machine learning. Statistics in Medicine, 29(3):337–346, 2010.
>
> - [9]: Hanna, J., Niekum, S. and Stone, P., 2019, May. Importance sampling policy evaluation with an estimated behavior policy. In International Conference on Machine Learning (pp. 2605-2613). PMLR.
>
> - [10]: Xie, Y., Zhu, Y., Cotton, C.A. and Wu, P., 2019. A model averaging approach for estimating propensity scores by optimizing balance. Statistical methods in medical research, 28(1), pp.84-101.
>
> - [11]: Tsiatis, Anastasios A. "Semiparametric theory and missing data." (2006): 206.
>
> - [12]: Shi, C., Song, R. and Lu, W., 2016. Robust learning for optimal treatment decision with np-dimensionality. Electronic journal of statistics, 10, p.2894.

---

> > ### Author Response · Authors · 2023-11-22
> > **Thanks for your time and consideration**
> >
> > Dear Reviewer p1oi,
> >
> > We would like to thank you again for your comments! We hope our response has addressed all your concerns. Since we are approaching the end of the discussion period, please let us know if you have any other questions, and we are happy to discuss more.
> >
> > Bests,
> > Authors

---

### Author Response · Authors · 2023-11-17
**Summary of responses and updates to the paper**

We thank all reviewers for their helpful comments. We applied the reviewer comments. The new changes in the new draft are in **blue** color. We also used **purple** color to highlight some discussions from the first submission. Here is a summary of our responses to the reviewer's comments and the main changes we made to the new draft:

 - Experiments based on **CIFAR-100** and **EMNIST** are added.

- Additional experiments for **CIFAR-10** are added.

 - We also carried out experiments on **KuaiRec** dataset as real-world dataset.

 - Introduction of a new upper bound on regret for our algorithms, detailed in our response to Reviewer G8No.

- Comprehensive discussion on partially known logging policy and propensity score estimation, as addressed in response to Reviewer p1oi.

- Comparison with pessimism-based methods in offline policy learning, elaborated in response to Reviewer CHNF.

- A discussion on assumptions in theoretical results is provided in response to Reviewer KCYk.

We hope that we have integrated feedback from all reviewers in the revised draft and answered their questions in the individual responses below. We are happy to discuss more and answer any additional questions the reviewers might have.

---

### Author Response · Authors · 2023-11-22
**Rebuttal summary**

We thank all reviewers for their helpful comments. Below is a summary outlining the **key recent changes** incorporated into our revised draft:

- In response to Reviewer p1oi comment, we conducted **additional experiments for CIFAR-10**, now available in Appendix F.6.
- In response to Reviewer p1oi comment, we included experiment results for two new datasets, **CIFAR-100 and EMNIST**, presented in Tables 7 and 9 in Appendix F.5.
- In response to Reviewer KCYk comment,  we added new experiments for **Q-learning** (specifically, reward function estimation) to Appendix H.1.
- In response to Reviewer G8No comment, we conducted a new experiment involving **Kuairec as a real-world dataset**, now detailed in Appendix F.7.
- In response to Reviewer G8No comment, we included a new theoretical result regarding the **regret bound** in Appendix H.3.
- In response to Reviewer CHNF comment, the discussion of **pessimistic methods** is added to Appendix A.
- In response to Reviewer CHNF comment, we added a discussion concerning the **effect of the logged missing-reward dataset** on the estimation of our regularizations (KL-divergence or reverse KL-divergence) in Appendix H.2.

We hope that we have integrated feedback from all reviewers in the revised draft.